# Categorical Flow Maps

**Daan Roos** [* 1] **Oscar Davis** [* 2] **Floor Eijkelboom** [* 1]
**Michael Bronstein** [2 3] **Max Welling** [1] **İsmail İlkan Ceylan** [4 3 2] **Luca Ambrogioni** [5] **Jan-Willem van de Meent** [1]

## Abstract

We introduce Categorical Flow Maps, a flow-matching method for accelerated few-step generation of categorical data via self-distillation. Building on recent variational formulations of flow matching and the broader trend towards accelerated inference in diffusion and flow-based models, we define a flow map towards the simplex that transports probability mass toward a predicted endpoint, yielding a parametrisation that naturally constrains model predictions. Since our trajectories are continuous rather than discrete, Categorical Flow Maps can be trained with existing distillation techniques, as well as a new objective based on endpoint consistency. This continuous formulation also automatically unlocks test-time inference: we can directly reuse existing guidance and reweighting techniques in the categorical setting to steer sampling toward downstream objectives. Empirically, we achieve state-of-the-art few-step results on images, molecular graphs, and text, with strong performance even in single-step generation.

## 1 Introduction

Consistency models (Song et al., 2023; Song & Dhariwal, 2023) have emerged as a powerful approach for accelerating inference in diffusion and flow-based generative models, enabling high-quality generation in one or a few steps. In the context of flow matching, Boffi et al. (2024; 2025) introduced a general framework for self-distillation through flow

maps, which not only learn the instantaneous velocity of the flow, but also the displacement along the flow's trajectories over any time interval.

Strong empirical results for few-step generation on images have been demonstrated using Mean Flows (Geng et al., 2025a;b) and Terminal Velocity Matching (Zhou et al., 2025b), which can be understood as Eulerian and Lagrangian formulations, respectively. While these efforts have focused on continuous domains, many practical applications involve inherently discrete structures, including text, sequences, and graphs. This naturally raises the question of whether such accelerated inference is also possible for discrete data.

To reason about self-distillation for discrete diffusion, it is helpful to distinguish two classes of underlying generative processes: those based on discrete interpolations and those based on continuous interpolations. Many recent advances have been based on discrete diffusion in which the forward and reverse processes are modelled as Markov chains, in either discrete (Austin et al., 2021; Sahoo et al., 2024; Lou et al., 2024; Shi et al., 2024) or continuous time (Campbell et al., 2022; Sun et al., 2023; Campbell et al., 2024). This line of work has led to results on scaling (von Rütte et al., 2025), competitive performance relative to strong autoregressive baselines (Nie et al., 2025), and principled methods for guidance (Li et al., 2024; Singhal et al., 2025; Schiff et al., 2025).

Yet, porting accelerated inference for methods based on discrete interpolations poses several challenges. One difficulty is that generation requires a large number of steps, with lower bounds quantifying this cost (Hayakawa et al., 2025; Kang et al., 2025), making these models resemble permuted autoregressive generators at test time. Moreover, flow map objectives cannot be applied directly, because of the discreteness and stochasticity of the trajectories, even when conditioned on both ends. Duo (Sahoo et al., 2025) partially side-steps this issue by using a continuous latent variable with $\arg\max$ decoding to induce deterministic paths, but it still operates on fundamentally discrete trajectories and employs a more restrictive distillation scheme than continuous-domain consistency methods.

In this work, we instead focus on continuous interpolations

---

[*]Equal contribution; ordering determined by a random draw
[1]UvA-Bosch Delta Lab, University of Amsterdam, Amsterdam, Netherlands [2]Department of Computer Science, University of Oxford, Oxford, UK [3]AITHYRA, Vienna, Austria [4]TU Wien, Vienna, Austria [5]Donders Institute for Brain, Cognition and Behaviour, Radboud University. Correspondence to: Oscar Davis <oscar.davis@cs.ox.ac.uk>, Floor Eijkelboom <f.eijkelboom@uva.nl>, Daan Roos <d.f.a.roos@uva.nl>.

*Proceedings of the $43^{rd}$ International Conference on Machine Learning*, Seoul, South Korea. PMLR 306, 2026. Copyright 2026 by the author(s).

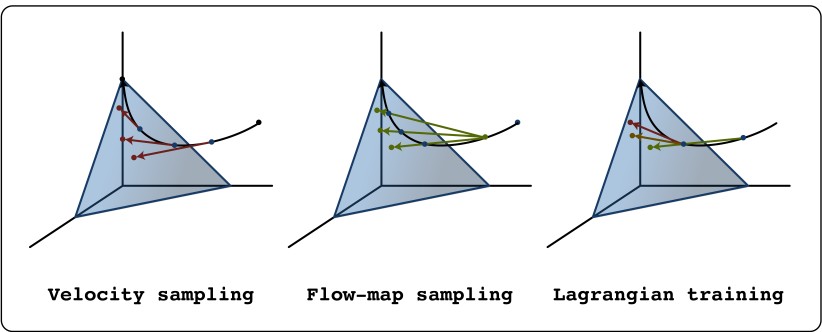 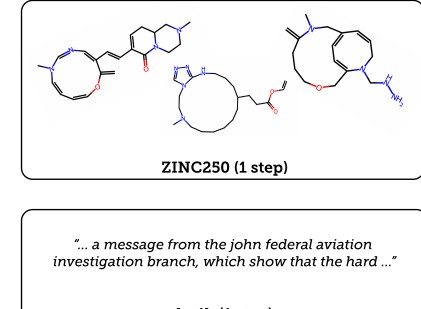

*Figure 1.* Overview of our method: CFM instantaneous velocity sampling (left), CFM flow-map sampling (middle), and Lagrangian training (right), plus their corresponding paths on a probability simplex. The red arrows denote the instantaneous velocity induced by $\pi_{t,t}$, the green arrows denote the flow map induced by $\pi_{s,t}$, and the yellow arrow denotes the time derivative of the flow map.

of discrete data (Dieleman et al., 2022; Davis et al., 2024; Eijkelboom et al., 2024; Cheng et al., 2025; Chen et al., 2023), where the continuous state space for trajectories makes consistency-style self-distillation techniques conceptually straightforward to apply.

We introduce *Categorical Flow Maps* (CFM), a simple flow matching construction for discrete data that learns a flow map transporting a *continuous* prior to a *discrete* target distribution. Inspired by variational flow matching (Eijkelboom et al., 2024), CFMs use a simplex-constrained endpoint parametrisation that not only enables compatibility with existing continuous-domain self-distillation methods, but also motivates a novel cross-entropy-based loss tailored to categorical data that provably bounds the Lagrangian residual. Empirically, we demonstrate high-quality few-step generation on discrete modalities. Specifically, we achieve with a single step over $95.8\%$ valid molecular graphs on QM9 and $93.5\%$ on ZINC, 10.1 FID on Binary MNIST, Text8 samples with 5.33 NLL, and LM1B with 142.7 Gen-PPL. An illustration is provided in Figure 1.

## 2 Background

In this work, we consider the typical generative modelling setup: given samples $x_1, \ldots, x_n \sim p_{\text{data}}$, we aim to learn a distribution, $\hat{p}_{\text{data}}$, such that $\hat{p}_{\text{data}} \approx p_{\text{data}}$, and which enables us to draw new samples.

### 2.1 Stochastic Interpolants

A broad class of state-of-the-art approaches (Lipman et al., 2023; Liu et al., 2023; Albergo et al., 2025) casts generation as the *dynamical transport of measures*: starting from a prior distribution $p_0$ that is easy to sample from, one constructs a dynamical system that pushes $p_0$ to the target distribution $p_1 \equiv p_{\text{data}}$. A stochastic interpolant is a stochastic process, $I : [0,1] \times \mathbb{R}^d \times \mathbb{R}^d \to \mathbb{R}^d$, that continuously interpolates

between $x_0 \sim p_0$ and $x_1 \sim p_1$, defined by

$$\forall t \in [0,1], \qquad I_t(x_0, x_1) := \alpha_t x_0 + \beta_t x_1, \qquad (1)$$

where $\alpha$ and $\beta$ are differentiable, and $I_0(x_0, x_1) = x_0$ and $I_1(x_0, x_1) = x_1$. We use a standard Gaussian base distribution $p_0 = \mathcal{N}(0, I)$ and draw $(x_0, x_1)$ from the independent coupling $p_0 \otimes p_{\text{data}}$. In this work, we consider the straight-line interpolation $\alpha_t \equiv 1 - t$ and $\beta_t \equiv t$ (see Appendix A.2 for the general case). Thus, $I_t$ induces a probability path, $(p_t)_{t \in [0,1]}$, that continuously moves from $p_0$ to $p_1$. Moreover, by its differentiability, the path also defines a probability flow, given by

$$\dot{x}_t = b_t(x_t), \qquad \text{with } x_0 \sim p_0, \qquad (2)$$

where the drift, $b$, is given by the conditional expectation

$$b_t(x_t) = \mathbb{E}[\dot{I}_t \mid I_t = x_t]. \qquad (3)$$

The velocity field is typically learned by minimising the following mean-squared error objective:

$$\mathcal{L}_{\text{fm}}(\theta) := \mathbb{E}_{t, x_0, x_1} \left[ \left\| v_t^\theta(x_t) - \dot{I}_t(x_t) \right\|^2 \right]. \qquad (4)$$

### 2.2 Flow Maps

Instead of learning the instantaneous vector field of the probability flow, one can learn integrals thereof to avoid, at inference time, the costly numerical integration (typically requiring tens to hundreds of network evaluations) required for high quality samples. This flow map framework was first introduced in Boffi et al. (2024; 2025).

We define the flow map, $X : [0,1]^2 \times \mathbb{R}^d \to \mathbb{R}^d$, for any two times $(s, t) \in [0,1]^2$, as the map which brings $x_s$ to $x_t$, for any solution $(x_t)_{t \in [0,1]}$ of (2); that is to say, $X_{s,t}(x_s) = x_t$. Knowing the flow map thus enables the

generation of samples from $p_1$ by simply drawing $x_0 \sim p_0$ and applying thereon $X_{0,1}$, yielding $X_{0,1}(x_0) = x_1 \sim p_1$. We parametrise these as

$$X_{s,t}(x_s) = x_s + (t-s)v_{s,t}(x_s), \quad (5)$$

so that the boundary condition $X_{t,t} = \text{Id}$ for any $t$ is automatically satisfied. Here, $v_{s,t}(x_s)$ represents the average velocity transporting $x_s$ from time $s$ to $t$; in particular, $v_{t,t}$ recovers the instantaneous velocity field $b_t$ from Equation (2).

Boffi et al. (2024; 2025) offer three different characterisations of flow maps, each of which gives rise to a training objective—we review these in more detail in Appendix A.1. In the context of our work, we focus on the Lagrangian self-distillation (LSD) objective, which offers the best empirical performance:

$$\mathcal{L}_{\text{LSD}}(\theta) := \mathbb{E} \left\| \partial_t X_{s,t}^\theta(x_s) - v_{t,t}^\theta(X_{s,t}(x_s)) \right\|^2,$$

where $t \sim \mathcal{U}(0,1)$, $s \mid t \sim \mathcal{U}(0,t)$, and $x_s = I_s(x_0, x_1)$. We use this convention for all expectations throughout. These losses are jointly optimised with a standard flow matching loss on $v_{t,t}^\theta$. A stop-gradient operator is applied to the $v_{t,t}^\theta$ term to stabilise training.

## 2.3 Variational Flow Matching

For $t \neq 1$, we can rewrite the drift as

$$b_t(x_t) = \frac{\mu_t(x_t) - x_t}{1-t}, \ \mu_t(x_t) := \mathbb{E}[x_1 \mid I_t = x_t]. \quad (6)$$

Since $b_t$ and $\mu_t$ determine each other, learning the velocity field is equivalent to estimating the conditional mean.

Variational Flow Matching (VFM) (Eijkelboom et al., 2024) formulates the learning problem in terms of an estimate for $\mu_t$ rather than $b_t$. Concretely, this estimate is defined as the mean of a variational family $q_t^\theta(x_1 \mid x_t)$. VFM then minimizes a time-averaged KL divergence to approximate the posterior $p_t(x_1 \mid x_t)$ of the probability path,

$$\mathbb{E} \left[ \text{KL}(p_t(x_1 \mid x_t) \| q_t^\theta(x_1 \mid x_t)) \right]. \quad (7)$$

Since the entropy of $p_t(x_1|x)$ does not depend on $\theta$, this is equivalent to minimising the negative log-likelihood

$$\mathcal{L}(\theta) = -\mathbb{E} \left[ \log q_t^\theta(x_1 \mid x_t) \right]. \quad (8)$$

Any variational family, the mean of which matches the true conditional mean $\mu_t$, recovers the correct velocity field via Equation (6). A simple and common choice is a mean-field factorisation across dimensions,

$$\log q_t^\theta(x_1 \mid x) := \sum_{d=1}^{D} \log q_t^\theta(x_1^{(d)} \mid x), \quad (9)$$

For continuous data, a Gaussian $q_t^\theta(x_1 \mid x) = \mathcal{N}(x_1 \mid \mu_t^\theta(x), \sigma^2 I)$ reduces the objective to a squared error on $\mu_t^\theta$, recovering the stochastic-interpolant regression loss. For discrete data, however, the variational perspective becomes essential: a categorical distribution $\text{Cat}(x_1 \mid \pi_t^\theta(x))$ over $K$ classes yields a cross-entropy objective, which keeps endpoint predictions on the simplex $\Delta^K := \{p \in \mathbb{R}^K : p_k \geq 0, \sum_{k=1}^{K} p_k = 1\}$ by construction. As we show next, this endpoint parametrisation also enables the use of flow-map self-distillation losses.

## 3 Categorical Self-Distillation

### 3.1 Constructing Categorical Flow Maps

As argued for previously, defining a generative process based on a continuous interpolation is crucial to leverage existing self-distillation losses. For this, we employ the VFM framework introduced in Section 2.3. However, VFM relies on endpoint predictions; that is, instead of learning the true vector field, $b_t$, we aim to learn $\mu_{1|t}$, the conditional expectation. It is thus necessary to port an endpoint prediction-based parametrisation for flow maps.

**Endpoint parametrisation.** Instead of the vector-based $v_{s,t}$ parametrisation, we opt for an endpoint one:

$$X_{s,t}(x_s) = x_s + (t-s)\frac{\pi_{s,t}(x_s) - x_s}{1-s}, \quad (10)$$

where $\pi : [0,1]^2 \times \mathbb{R}^d \to \Omega$, and $\Omega$ is the support of the data. The "partial denoiser" $\pi_{s,t}(x_s)$ is endpoint-valued: it predicts a point in the data support, not the intermediate state $x_t$. The flow map then moves from $x_s$ only a fraction $(t-s)/(1-s)$ toward this *intermediate* predicted endpoint. This choice maintains two crucial properties of flow maps:

$$\lim_{s \to t} \partial_t X_{s,t}(x_s) = \frac{\pi_{t,t}(x_t) - x_t}{1-t} = b_t(x_t) \quad (11)$$

$$(1-t)\partial_t X_{s,t}(x_s) = \pi_{t,t}(X_{s,t}(x_s)) - X_{s,t}(x_s). \quad (12)$$

The first one is the "tangent condition": it is crucial as it shows that, at times $s = t$, our parameterisation still retrieves the instantaneous vector field of the probability flow. The second one is the Lagrangian condition, which enables self-distillation training. The proofs are in Appendix B.1.

From this discussion, it is easy to see that we can apply flow map objectives on $s < t$ times, alongside the VFM endpoint loss for $s = t$, solely relying on a partial denoiser supported on the simplex, $\Omega = \Delta^K$. This shows that it is possible to train a flow map using an endpoint parametrisation, which allows us to leverage the inductive bias given by the knowledge of $\Omega$.

**Training objective.** Parametrising our learnt flow map, $X^\theta$, as in (10) (resp., with $\pi^\theta$), we define the VFM part of our

loss as in Equation (8):

$$\mathcal{L}_{\inf}(\theta) := -\mathbb{E}\left[\log q_{t,t}^\theta(x_1 \mid x_t)\right], \quad (13)$$

where $q_{t,t}^\theta(x_1 \mid x_t) := \prod_{d=1}^D \mathrm{Cat}(x_1^d \mid \pi_{t,t}^{\theta,d}(x))$, as before. In implementation, the network outputs logits $\ell_{s,t}^\theta(x_s) \in \mathbb{R}^{D \times K}$, and we set

$$\pi_{s,t}^\theta(x_s) = \mathrm{softmax}_K(\ell_{s,t}^\theta(x_s)),$$

with the softmax applied over the categorical dimension. From Equation (12), we can also define our Lagrangian self-distillation as:

$$\mathcal{L}_{\mathrm{CSD}}(\theta) := \mathbb{E}\left[w_t \left\| r_{s,t}^\theta(X_s) \right\|^2\right] \quad (14)$$

where $w_t$ is a time-dependent weight, and

$$r_{s,t}^\theta(x) := (1-t)\partial_t X_{s,t}^\theta(x) - \pi_{t,t}^\theta(X_{s,t}^\theta(x)) + X_{s,t}^\theta(x).$$

We found that the typical choice of $w_t \equiv (1-t)^{-2}$ was too unstable near time one (despite clamping), while $w_t \equiv (1-t)^{-1}$ performed better, but simply $w_t \equiv 1$ had the best performance and the most stable training.

Finally, as a direct consequence of Boffi et al. (2025), the true flow map is a minimiser of (14), and uniqueness is given by some mild regularity assumptions on the true flow. We combine the two losses as $\mathcal{L} = \mathcal{L}_{\inf}(\theta) + \lambda \mathcal{L}_{\mathrm{CSD}}(\theta)$, with $\lambda = 1$ unless stated otherwise (see Appendix E.2.1 for an ablation).

## 3.2 Distillation through Endpoint Consistency

While the usual Lagrangian self-distillation loss is satisfactory in theory, in practice, there may be a mismatch between its magnitude and that of the VFM loss. Indeed, the former is a mean-squared error loss, and the latter a cross-entropy one, sometimes leading to differences of up to two orders of magnitude, in practice. This motivates an endpoint-consistency objective that distills categorical endpoint predictions while retaining an explicit temporal-drift penalty. We show that such an objective is achievable through the following *Endpoint-Consistent Lagrangian Distillation* (ECLD) loss bound, proven in Appendix B.2.

---

**Proposition 3.1** (ECLD controls the Lagrangian residual)**.** *Under the linear VFM decoder and the induced flow map from* (10), *define the endpoint consistency loss*

$$\mathcal{L}_{\mathrm{EC}}(\theta) := \mathbb{E}\left[w_t \, \mathrm{KL}(\pi_{t,t}^\theta(X_{s,t}(x_s) \| \pi_{s,t}^\theta(x_s))\right], \quad (15)$$

*with $w_t = (1-t)^{-2}$, and the temporal drift term*

$$\mathcal{L}_{\mathrm{TD}}(\theta) := \mathbb{E}\left[\gamma_{s,t}^2 \, \left\| \partial_t \pi_{s,t}^\theta(x_s) \right\|_2^2\right], \quad (16)$$

---

*where $\gamma_{s,t} = (t-s)/(1-s)$. Then*

$$\mathcal{L}_{\mathrm{CSD}} \leq 4\mathcal{L}_{\mathrm{EC}} + 2\mathcal{L}_{\mathrm{TD}}. \quad (17)$$

---

The bound decomposes the Lagrangian residual into two terms. The first measures *endpoint consistency*; i.e. whether the "teacher" prediction (the instantaneous vector field with a stop-gradient) at the transported state, $X_{s,t}(x_s)$, agrees with the student's prediction, $\pi_{s,t}(x_s)$, that generated the transport. The second is a drift term that arises because the student's endpoint predictor $\pi_{s,t}$ varies with $t$. When both terms vanish—making the model self-consistent and the endpoint predictor time-stable—the Lagrangian residual is zero and the learnt map is the unique flow map.

In practice, we minimise a cross-entropy variant rather than KL divergence. Indeed, since $\mathrm{CE}(p,q) = \mathrm{KL}(p\|q) + H(p)$ and entropy is non-negative, the bound remains valid. Because of the stop-gradient operator a simple cross-entropy loss yields the same gradients. This yields a self-distillation objective that inherits the same Lagrangian guarantees as well:

---

**Corollary 3.2** (Cross-entropy ECLD)**.** *Let $\pi_{s,t}^{\mathrm{tgt}}(x_s) := \mathrm{sg}\left(\pi_{t,t}^\theta(X_{s,t}(x_s))\right)$, and define*

$$\mathcal{L}_{\mathrm{CE\text{-}EC}}(\theta) := \mathbb{E}\left[w_t \mathrm{CE}\left(\pi_{s,t}^{\mathrm{tgt}}(x_s), \pi_{s,t}^\theta(x_s)\right)\right]. \quad (18)$$

*Then*

$$\mathcal{L}_{\mathrm{CSD}} \leq 4\mathcal{L}_{\mathrm{CE\text{-}EC}} + 2\mathcal{L}_{\mathrm{TD}}. \quad (19)$$

---

Hereinafter, we refer to the combined objective $\mathcal{L}_{\mathrm{ECLD}} := 4\mathcal{L}_{\mathrm{CE-EC}} + 2\mathcal{L}_{\mathrm{TD}}$ as the ECLD loss. Notably, both terms operate on endpoint predictions rather than velocities. The consistency term uses cross-entropy too, which is directly interpretable on the simplex; the temporal drift term regularises how the endpoint predictor varies with target time.

*Remark* 3.3. While one could apply flow-map objectives by learning unconstrained velocity fields directly, this ignores the simplex structure of categorical endpoints. Our endpoint parametrisation instead predicts a categorical endpoint and moves toward it, keeping the learned map tied to the geometry of the data support. Empirically, the comparison to Naive Flow Maps in Section 4.1 suggests that this inductive bias becomes more important in higher-dimensional settings, with larger gains on ZINC than on QM9. We discuss this comparison further in Appendix C.

## 3.3 Test-Time Inference in Categorical Flow Maps

A practical advantage of formulating categorical flow maps fully within the stochastic-interpolant (SI) framework is that existing SI test-time inference techniques carry over directly. As described in Sabour et al. (2025a), given a current state $X_{t,1}$, the learned flow map provides a cheap and accurate

lookahead of the end of the trajectory, which we can use to steer sampling towards a downstream objective *at test time*.

Letting, for any $t \neq 1$, $v_{t,t}(x_t) = \frac{\pi_{t,t}(x_t) - x_t}{1-t}$, we can sample stochastically from the learned dynamics of our flow, by compensating for the added noise term in our score:

$$\mathrm{d}x_t = \left[ v_{t,t}(x_t) + \frac{\sigma_t^2}{2} s_t(x_t) \right] \mathrm{d}t + \sigma_t \, \mathrm{d}W_t, \qquad (20)$$

where $s_t(x_t) = (t v_{t,t}(x_t) - x_t)(1-t)^{-1}$ denotes the score, $\sigma : \mathbb{R} \to \mathbb{R}$ is the diffusion coefficient, and $(W_t)_{t \in [0,1]}$ is a standard Wiener process. Using $X_{t,1}$ to infer the endpoint of the current trajectory accurately, we can use any differentiable reward $r(x)$ to adjust the above score to sample from a reward-tilted distribution $\tilde{p}(x) := p(x) e^{r(x)+F}$, where $F$ is a normalization factor, as follows:

$$\tilde{s}_t(x_t) = s_t(x_t) + t \nabla_{x_t} r(X_{t,1}(x_t)), \qquad (21)$$

Here $r(x) = \log p(y \mid x)$ can be a classifier trained on clean data. This *lookahead* can also be done by using the $\mu_{t,t}$, or the reward can be applied on the noisy state directly, which we will compare in the results.

As outlined in Sabour et al. (2025a), the above dynamics require a correction to sample from the tilted distribution in an unbiased way. To this end, we use sequential Monte-Carlo (SMC) sampling during generation, as the above formulation fails to match the desired target distribution exactly. It uses importance weights to perform resampling—deleting low-reward trajectories and duplicating high-reward ones—which stabilises the sampling process and focuses computational effort on regions that satisfy the problem constraints. This mechanism ensures the final output is theoretically grounded and accurately tracks the reward-tilted distribution without requiring expensive retraining. In our experiments, we reuse the SMC settings from Sabour et al. (2025a).

In our setting, the reward model is trained on discrete objects (*e.g.*, one-hot/categorical or binary variables), while the Flow Map naturally evolves in a continuous relaxation, producing soft states in the probability simplex. We can therefore evaluate $r$ and its gradients using one of two standard choices: (i) apply a straight-through (STE) discretization to $X_{t,1}(x_t)$ so that the reward sees a hard discrete sample in the forward pass, while gradients are propagated through the underlying soft state; or (ii) feed the soft simplex-valued prediction directly into the reward model (or a reward model trained with such soft inputs). Both are valid test-time inference strategies in our framework; in our experiments, STE consistently yielded stronger guidance.

## 4 Experiments

We evaluate Categorical Flow Maps on three discrete data modalities: molecular graphs (QM9, ZINC250k), binarised

images (MNIST), and text (Text8, LM1B). Throughout, we learn a partial denoiser by parametrising its logits, retrieving its actual prediction at inference by applying a softmax.

### 4.1 Graph Generation

We evaluate categorical flow maps on molecular graph generation using QM9 (Wu et al., 2017) (up to 9 heavy atoms) and ZINC250k (Sterling & Irwin, 2015) (up to 38 heavy atoms), comparing against multi-step diffusion and flow baselines (GDSS (Jo et al., 2022), GruM (Jo et al., 2024), CatFlow (Eijkelboom et al., 2024), DeFoG (Qin et al., 2025)) as well as one-shot and few-step methods (Set2GraphVAE (Vignac & Frossard, 2022), MoFlow (Zang & Wang, 2020), PairFlow (Park et al., 2025)).

We follow the standard molecular graph generation setting and generate atom types and bond types, not 3D atomic coordinates. Our goal is not to surpass multi-step diffusion and flow baselines, but to match their sample quality with far fewer function evaluations. To isolate the effect of the endpoint parametrisation, we train an otherwise identical model using the standard Lagrangian loss with unconstrained velocities (Naive Flow Map). Full experimental details are provided in Appendix E.

**Results.** Table 1 compares sampling quality as a function of the number of function evaluations (NFE) on the QM9 dataset. Among single-step generators, our method substantially improves validity over prior one-step baselines (Set2GraphVAE, MoFlow) and over PairFlow, while maintaining high uniqueness. At two steps, our sampler reaches near-saturated validity ($91 - 97\%$) and high uniqueness ($96 - 97\%$), while achieving an FCD competitive with strong multi-step flow-based baselines. Notably, while unconstrained Flow Maps achieve reasonably competitive performance on QM9, the benefits of our simplex-constrained endpoint parametrisation become substantially more apparent on the larger, more complex molecules of ZINC.

On ZINC, few-step generation is considerably more challenging due to the larger molecular structures. Nevertheless, our methods substantially outperform one-shot baselines: ECLD achieves 93.5% validity at 1 NFE, far exceeding MoFlow (63.1%) and PairFlow (11.2%). At two steps, both CSD and ECLD achieve FCD scores ($12.1 - 12.2$) that match or improve upon multi-step baselines such as CatFlow (13.2 at 100 NFE) and GDSS (14.7 at 1,000 NFE), representing a $50 - 500\times$ reduction in function evaluations.

Figure 2 visualises the quality–NFE trade-off across both datasets. Between our two losses, ECLD tends to favour validity while CSD achieves lower FCD at matched step counts. Notably, both CSD and ECLD outperform standard (unconstrained) Flow Maps in terms of validity and FCD, demonstrating the benefit of the simplex-constrained

| Method | NFE | QM9 | | | ZINC | | |
|---|---|---|---|---|---|---|---|
| | | Valid ↑ | Unique ↑ | FCD ↓ | Valid ↑ | Unique ↑ | FCD ↓ |
| *Multi-Step Diffusion / Flow (reference)* | | | | | | | |
| GDSS (Jo et al., 2022) | 1,000 | 95.7 | 98.5 | 2.90 | 97.0 | 99.6 | 14.7 |
| GruM (Jo et al., 2024) | 1,000 | 99.7 | – | **0.11** | 98.7 | – | 2.26 |
| CatFlow (Eijkelboom et al., 2024) | 100 | **99.8** | **99.9** | 0.44 | **99.2** | **100.0** | 13.2 |
| DeFoG (Qin et al., 2025) | 500 | 99.4 | 96.3 | 0.12 | **99.2** | 99.9 | **1.45** |
| DeFoG (Qin et al., 2025) | 50 | 99.2 | 96.2 | 0.26 | 96.7 | 99.9 | 2.12 |
| *Few-Step* | | | | | | | |
| Set2GraphVAE (Vignac & Frossard, 2022) | 1 | 59.9 | 93.8 | – | – | – | – |
| MoFlow (Zang & Wang, 2020) | 1 | 91.4 | 98.7 | 4.46 | 63.1 | 99.9 | 20.9 |
| PairFlow (*) (Park et al., 2025) | 1 | 44.3 | **99.5** | – | 11.2 | **100.0** | – |
| PairFlow (*) (Park et al., 2025) | 2 | 66.9 | **99.4** | – | 42.6 | **100.0** | – |
| Naive Flow Map (Boffi et al., 2025) | 1 | 86.0 | 91.6 | 2.50 | 59.5 | 99.9 | 21.7 |
| Naive Flow Map (Boffi et al., 2025) | 2 | 89.3 | 97.6 | 0.70 | 51.6 | **100.0** | 16.1 |
| CFM CSD (ours) | 1 | 89.8 | 92.2 | **2.07** | 73.5 | 99.9 | **19.3** |
| CFM CSD (ours) | 2 | 91.2 | 97.7 | **0.44** | 67.0 | 99.9 | 12.1 |
| CFM ECLD (ours) | 1 | **96.3** | 88.2 | 2.19 | **93.5** | 99.5 | 20.0 |
| CFM ECLD (ours) | 2 | **96.2** | 97.6 | 0.59 | **82.4** | 99.9 | **12.2** |

*Table 1.* Molecular generation results on QM9 and ZINC. Top shaded rows indicate multi-step reference methods; our goal is to match their quality with far fewer sampling steps. A dash indicates that the metric was not reported by the corresponding source. In each sub-category (and for each NFE for few-step models), we make bold the best result. (*) PairFlow numbers are taken from the original paper and were reported using $N = 1,024$ generated samples; see Table 3 for a matched comparison. MoFlow results from Jo et al. (2024).

endpoint parametrisation. An ablation over the CSD distillation weight $\lambda$ (Appendix E.2.1) reveals that this gap is not intrinsic to the objective. Increasing $\lambda$ from 1 to 25 substantially improves CSD's one-step validity and FCD, matching or exceeding ECLD in the low-NFE regime. However, larger $\lambda$ degrades multi-step quality, where the default $\lambda = 1$ achieves the best FCD at 2 and 5 steps. ECLD avoids this trade-off, yielding strong one-step performance without tuning. In practice, ECLD or CSD with tuned $\lambda$ are both viable for single-step generation, while CSD at $\lambda = 1$ is preferable when a multi-step budget is available. In Figure 6 we extend the results of Figure 2 ($\lambda = 1$) to NFEs up to 100. The CSD model continues to improve more steeply with additional NFE compared to Naive Flow Matching and ECLD, which level off earlier.

Figure 8 compares Flow Map sampling with Euler integration of the learned instantaneous velocity field. For the Naive and CSD losses, Flow Map sampling consistently outperforms Euler integration at matched step counts. For ECLD, however, Euler integration achieves comparable or better FCD at higher NFE, suggesting that the ECLD-trained flow map is most accurate in the low-NFE regime. Interestingly, the instantaneous velocity field learned with CSD also outperforms that of ECLD at higher NFE, despite both losses using the same $\mathcal{L}_{\text{inf}}$ term for the instantaneous velocity. This is consistent with the broader pattern observed

in the $\lambda$ ablation (Section E.2.1): CSD benefits more from additional function evaluations, while ECLD concentrates its accuracy in the few-step regime. In practice, one can use flow map sampling for 1–5 steps and switch to Euler integration when additional NFE is available.

### 4.2 Image Generation, Test-Time Guidance

We study unconditional *image* generation on binarised MNIST (Deng, 2012) digits ($28 \times 28$), viewed as a grid of binary categorical variables (foreground/background). We use this simple visual domain as a controlled test-bed to analyse sample quality, especially for *test-time guidance*, in our categorical flow-map framework. The goal is twofold: (i) achieve high-quality *few-step* sampling (small NFE) for image synthesis, and (ii) study whether test-time guidance through reward tilting also applies to the discrete domain. We define a reward given by $R(x) \coloneqq \log p(y = 0 \mid x)$, aiming to tilt the samples towards zeroes, using a simple classifier. For steering, we compare 1) flow-map lookahead, 2) applying the reward to the 'instantaneous' inferred state $\pi_{t,t}$ (denoiser lookahead), and 3) applying the reward on the noisy state directly (no lookahead). For the experimental setup, we directly follow Park et al. (2025). All results are averaged over 10 runs.

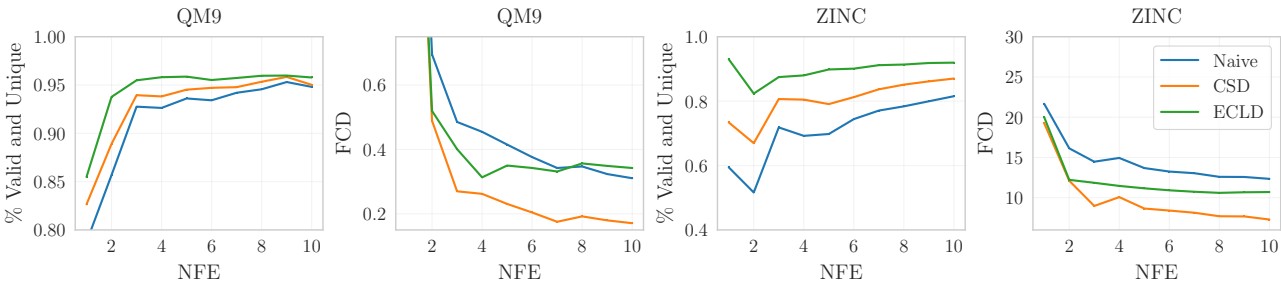

*Figure 2.* Sample quality versus NFEs on QM9 (two leftmost panels) and ZINC (two rightmost panels), showing the fractions of valid and unique molecules (↑), and FCD (↓). We compare our methods (CSD, ECLD) against the unconstrained Flow Map baseline (Naive).

### 4.3 Text Generation

**Datasets.** We train CFMs over the Text8 (Mahoney, 2006) and LM1B (Chelba et al., 2014) datasets. Text8 is a 100MB dataset of Wikipedia data in English, and we follow the data setup of Campbell et al. (2024): we tokenize the text with 27 tokens for each letter, and a space; numbers are spelled out (*e.g.*, 1 becomes "one"); and we model sequences long of 256 tokens at once. LM1B is a dataset composed of one billion words in English, for which we set up the data following Sahoo et al. (2025): we tokenize the sequences with the `bert-base-uncased` tokenizer, containing about 30K tokens, with sequences of 128 tokens long, and using sequence packing. For the evaluation of our NLLs and Gen-PPLs, we sample 512 samples from each model respectively, and report the average obtained over these.

**Baselines.** The closest comparable baseline available to the best of our knowledge is Duo (Sahoo et al., 2025), for which checkpoints and Generative-Perplexity (Gen-PPL) results are available for the OpenWebText (OWT) (Gokaslan & Cohen, 2019) dataset only—which is computationally much more demanding. We therefore compare against DFM (Campbell et al., 2024) for Text8, and against Duo, which we train ourselves, for LM1B, respectively. We use the Gen-PPL of a large pre-trained model: GPT-J-6B (Wang & Komatsuzaki, 2021) for Text8 and `gpt2-large` (Radford et al., 2019), as in the aforementioned papers.

**Architecture.** A crucial aspect of the text experiments is the underlying neural network architecture. While we base our model based on that of Lou et al. (2024), as in Sahoo et al. (2025; 2024), we found that several improvements were required. First of all, we included the latest JVP kernel for forward automatic differentiation of fast attention kernels, developed in Zhou et al. (2025b), which we found to reduce the memory footprint drastically (by up to approx. 50%). We also found that rewriting the architecture to have better compatibility with compilation features of PyTorch to allow for some noticeable speed-ups (up to 10% in iterations/sec). Finally, we found that changing the embedding layer was of crucial importance, as the softmax applied in Sahoo et al.

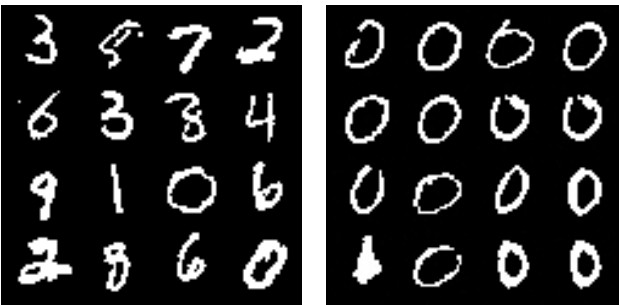

*Figure 3.* Samples from our categorical flow map on binary MNIST: (left) unconditional (right) and tilted by the classifier reward towards zeroes.

| Guidance Method | FID ↓ | Accuracy ↑ |
|---|---|---|
| Flow Map Lookahead | **36.9** | **84.7** |
| Denoiser Lookahead | 38.7 | 82.5 |
| No Lookahead | 40.7 | 83.1 |

*Table 2.* Test-time guidance results on binarised MNIST for generating digit "0". We compare three guidance strategies: using the flow map $X_{t,1}$ for lookahead, using the instantaneous denoiser $\pi_{t,t}$, and standard guidance without lookahead.

**Results.** Compared to Park et al. (2025), we obtain improved FID in few-step sampling. We report **10.1** vs **12.9** for one step, **8.7** vs **9.3** for two steps, and **7.8** vs **8.5** for four steps. This shows indeed that categorical flow maps can obtain high-quality sampling with few NFEs. Moreover, as shown in Figure 3, test-time reward steering successfully concentrates samples on the target digit class. Table 2 compares three guidance strategies for steering generation toward a target digit class. Using the flow map for lookahead achieves the best class-conditional FID, indicating higher sample quality, while also obtaining the highest target-class accuracy (*i.e.*, reward satisfaction). This demonstrates that categorical flow maps with straight-through estimation of gradients enabling effective test-time guidance for discrete data.

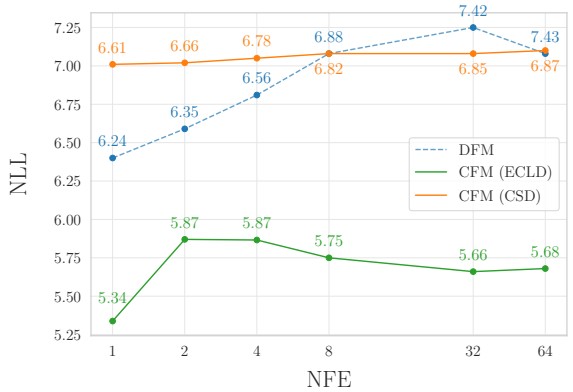

*Figure 4.* Text8: NFE against NLL as measured by GPT-J-6B. The numbers next to the points are the corresponding entropies on the token space of the GPT model.

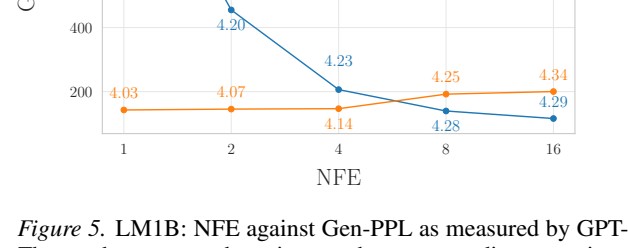

*Figure 5.* LM1B: NFE against Gen-PPL as measured by GPT-2. The numbers next to the points are the corresponding entropies on the token space of the GPT model.

(2025) caused our model to severely underfit. We discuss our choices for that layer in Appendix F.1. Another crucial aspect was to implement the Lipschitz-controlling tricks from Zhou et al. (2025b) for stable training. For Text8, our model contains 100M parameters, and 140M for LM1B.

**Results.** Our results are available in Figure 4 for Text8, and in Figure 5 for LM1B. For Text8, we can see that the NLL offered by ECLD outperforms that of DFM for all NFEs, while keeping a relatively close entropy, given the high dimensionality of the token space. As for LM1B, we first note that the Gen-PPL is naturally higher, because of the difference in distribution with the training set of GPT-2, which is closer to OWT.[1] Secondly, to ensure a fair comparison, we evaluate both models using the exact same evaluation pipeline, based on the publicly available code from Sahoo et al. (2025). Under these identical conditions, CFM outperforms Duo (Sahoo et al., 2025) in the low NFE regime, with up to $7.4\times$ lower PPL for 1 NFE. While the PPL metric may be highly volatile and thus strict monotonicity may not hold, we observe that our method's performance does not increase with more steps with these settings, and Duo exceeds our model's performance at 8 and 16 NFE. We provide some qualitative samples for our one-step generation in Appendix F.3.

## 5  Related Works

**Discrete diffusion.** First defined in its current and general formulation in Austin et al. (2021), discrete diffusion noises a data distribution, $p_{\text{data}} \equiv p_0$, in time to a prior distribution, $p_1$, independent of $p_0$, defining a probability path over categorical distributions, $(p_t)_{t\in[0,1]}$. It is then possible to simulate the reverse continuous-time Markov

chain (CTMC), bringing $p_1$ back to $p_0$, knowing the distribution of $p_{0|t}$. When the prior is masked ($p_1 = \delta_{\mathbf{e}_M}$), the method is referred to as masked diffusion and the objective reduces to a simple cross-entropy loss (Sahoo et al., 2024; Shi et al., 2024). Similarly, for a masked prior, one can instead learn the reverse transition rate matrix (Campbell et al., 2024), or the discrete vector field from a flow matching formulation (Gat et al., 2024)—both essentially reducing to cross-entropy-based losses—to reverse the same CTMC (or simulate it forward in flow matching). When the prior is uniform ($p_1 = \mathbf{1}/K$, for $K > 1$ categories) (Schiff et al., 2025), a more involved objective is available. SEDD (Lou et al., 2024) proposes a score-based perspective, solidifying the concrete score generalisation from Meng et al. (2023).

**Continuously-parametrised discrete diffusion.** Instead of simulating discrete Markov chains, a host of works considered evolving a flow on a continuous space, typically on $\mathbb{R}^K$ and one-hot encoding the data points, the target distribution, $p_1$, remaining discrete. Stark et al. (2024) claim that linear paths supported on the simplex are limited and thus construct paths based on the Dirichlet distributions, but therefore introduces terms unstable in higher dimensions and fails to scale. Davis et al. (2024); Cheng et al. (2025) endow the simplex with the Fisher-Rao metric and leverage the isometry to the scaled positive orthant of the sphere, which remains more stable in high dimensions. Dunn & Koes (2024), similarly to our work, completely discard the simplex structure, and use a Gaussian prior, but their endpoint-based training uses an MSE objective, hence underperforming our more principled method and lacking self-distillation from scratch methods. In general, the aforementioned methods underperform their fully discrete counterparts.

**Accelerated methods.** While diffusion and flow matching offered a leap in performance on continuous data, their inference is typically expensive, requiring many inference steps

---

[1]A random batch without sequence packing obtains a Gen-PPL of about 100.

for high quality generation. Many works since then have established ways of training *from scratch* models that generate high quality samples in as few as one or two steps (Song et al., 2023; Song & Dhariwal, 2023; Lu & Song, 2025; Geng et al., 2025a; Zhou et al., 2025a; Sabour et al., 2025b; Boffi et al., 2025; Kim et al., 2024).

For discrete diffusion, by default, it is hard to construct a consistency-like training objective, since the noisy samples are, at all times $t$, *sampled* from $p_{t|0,1}$, introducing stochasticity, especially near time *one*. Moreover, at inference, each step samples a subset of the elements of the sequence *independently*, provably leading to low quality generation (Hayakawa et al., 2025; Kang et al., 2025). Sahoo et al. (2025) show how to partially eliminate the stochasticity to leverage an algorithm similar to SDTT (Deschenaux & Gulcehre, 2025), but generation remains challenging in the very low-step regime, and the distillation requires a second, separate stage. ReDi (Yoo et al., 2025) consider refining couplings for accelerated sampling. Zhu et al. (2025) show how to create a one-step sampler for masked diffusion. Finally, Xu et al. (2025) also show that, by correcting for the independence assumption of discrete diffusion through an energy-based model, the number of required sampling steps can be reduced.

## 6 Conclusion

We introduced *Categorical Flow Maps*, a self-distillable flow-based framework for accelerated generation of discrete data. CFMs build on a variational, endpoint-prediction view of flow matching, and learn a *flow map* that transports a simple *continuous* prior to a *discrete* target while remaining constrained to the probability simplex. This endpoint parametrisation yields a likelihood-aligned learning signal (cross-entropy for categorical endpoints) and, crucially, makes flow-map self-distillation applicable in discrete domains. Moreover, beyond the standard Lagrangian objective, we derived an *endpoint-consistency* self-distillation objective (ECLD), which controls the Lagrangian residual while operating directly on simplex-valued endpoint predictions.

Empirically, CFMs deliver strong few-step generation performance across multiple categorical generative tasks, and we further demonstrate that our continuous flow map formulation enables natural test-time steering via standard guidance and reweighting machinery adapted to simplex-valued trajectories. Overall, our work provides a practical route to scalable, self-distillable generative modelling for discrete data, bridging the gap between accelerated continuous-domain flow maps and categorical generation.

## Acknowledgements

OD is funded by both Project CETI and Intel. FE, DR, and JWvdM acknowledge the Bosch Center for Artificial Intelligence. This research is partially supported by the EPSRC Turing AI World-Leading Research Fellowship No. EP/X040062/1 and EPSRC AI Hub No. EP/Y028872/1. This work used the Dutch national e-infrastructure with the support of the SURF Cooperative using grant no. EINF-17357. Computations were partially performed using the University of Amsterdam - Science Faculty (UvA/FNWI) HPC Facility.

## Impact Statement

This paper presents work whose goal is to advance the field of Machine Learning. There are many potential societal consequences of our work, none which we feel must be specifically highlighted here.

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

# A    Additional Background

## A.1    Flow Map Matching via Self-Distillation

We give a brief review of flow map matching learning via self-distillation, summarising the work of Boffi et al. (2025). Recall that a flow map is the map, $X : [0,1]^2 \times \mathbb{R}^d \to \mathbb{R}^d$, which, for any solution of the probability flow, and any two times $(s,t) \in [0,1]^2$, satisfies the *jump condition*: $X_{s,t}(x_s) = x_t$. We can easily deduce the three characterisations from Boffi et al. (2025) from this simple property.

**Lagrangian condition.** To derive the Lagrangian condition, we can first show the so-called "tangent condition". From the vector field parametrisation, we have that

$$x_t = X_{s,t}(x_s) = x_s + (t-s)v_{s,t}(x_s) \implies \frac{x_t - x_s}{t-s} = v_{s,t}(x_s), \tag{22}$$

for any times $s \neq t$. From the limit as $s$ tends to $t$, it then becomes clear that $\partial_t x_t = b_t(x_t) = v_{t,t}(x_t)$. Then, since $X_{s,t}(x_s) = x_t$, by differentiating w.r.t. $t$, we also get that $\partial_t X_{s,t}(x_s) = b_t(x_t) = v_{t,t}(x_t) = v_{t,t}(X_{s,t}(x_s))$. Thus arises Lagrangian self-distillation (LSD):

$$\mathcal{L}_{\mathrm{LSD}}(\theta) := \mathbb{E} \left\| \partial_t X_{s,t}^\theta(x_s) - v_{t,t}^\theta(X_{s,t}(x_s)) \right\|^2. \tag{23}$$

Boffi et al. (2025); Davis et al. (2025) recommend placing a stop-gradient operator on $v_{t,t}^\theta(X_{s,t}^\theta(x_s))$.

**Eulerian condition.** We have that, for any times $s$ and $t$, $X_{t,s} \circ X_{s,t} = \mathrm{Id}$. It suffices then to take the partial derivative with respect to $s$ of this relation to find that the flow map satisfies

$$\partial_s X_{s,t}(x_s) + \nabla X_{s,t}(x_s)v_{s,s} = 0. \tag{24}$$

This naturally gives rise to the Eulerian self-distillation (ESD) loss:

$$\mathcal{L}_{\mathrm{ESD}}(\theta) := \mathbb{E} \left\| \partial_s X_{s,t}^\theta(x_s) + \nabla X_{s,t}^\theta(x_s)v_{s,s}^\theta(x_s) \right\|^2. \tag{25}$$

Following Boffi et al. (2025); Davis et al. (2025), a stop-gradient operator should be placed on the Jacobian term.

**Semigroup condition.** The flow map has a semigroup structure in time: for $(s,u,t) \in [0,1]^3$, it is easy to see that

$$X_{u,t} \circ X_{s,u} = X_{s,t}, \tag{26}$$

whence the "Progressive" self-distillation (PSD) optimisation objective:

$$\mathcal{L}_{\mathrm{PSD}}(\theta) := \mathbb{E} \left\| X_{s,t}^\theta(x_s) - X_{u,t}^\theta(X_{s,u}^\theta(x_s)) \right\|^2. \tag{27}$$

Once more, according to Boffi et al. (2025); Davis et al. (2025), a stop-gradient operator should be placed on the two-step term.

## A.2    General Affine Interpolation

The straight-line interpolation $x_t = tx_1 + (1-t)x_0$ is a specific case of the more general family of affine interpolations

$$I_t(x_0, x_1) := \alpha_t x_0 + \beta_t x_1, \tag{28}$$

where $\alpha_t$ and $\beta_t$ are differentiable scalar schedules (chosen such that $X_t$ interpolates from $X_0$ to $X_1$ as $t$ goes from 0 to 1).

The marginal velocity field, the conditional expectation of the pathwise time derivative, is then given by

$$b_t(x) := \mathbb{E}\left[\dot{\alpha}_t x_0 + \dot{\beta}_t x_1 \mid I_t = x_t\right]. \tag{29}$$

In this more general case, the conditional expectation is taken under the posterior probability path $p_t(x_0, x_1 | x_t)$. The derivative of the affine interpolant gives the pathwise derivative

$$\partial_t x_t = \dot{\alpha}_t x_0 + \dot{\beta}_t x_1 = \frac{\dot{\alpha}_t}{\alpha_t} x_t + \left(\dot{\beta}_t - \beta_t \frac{\dot{\alpha}_t}{\alpha_t}\right) x_1, \tag{30}$$

where we substituted $x_0 = (x_t - \beta_t x_1)/\alpha_t$. Taking conditional expectations given $I_t = x_t$ gives a convenient representation of the marginal velocity field:

$$b_t(x_t) = \frac{\dot{\alpha}_t}{\alpha_t} x_t + \left( \dot{\beta}_t - \beta_t \frac{\dot{\alpha}_t}{\alpha_t} \right) \mu_{1|t}(x_t), \tag{31}$$

where we define

$$\mu_{1|t}(x_t) := \mathbb{E}[X_1 \mid I_t = x_t]. \tag{32}$$

In the general affine case we parametrise the average velocity as

$$v_{s,t}^{\theta}(x_s) := \frac{\dot{\alpha}_s}{\alpha_s} x_s + \left( \dot{\beta}_s - \beta_s \frac{\dot{\alpha}_s}{\alpha_s} \right) \pi_{s,t}^{\theta}(x_s). \tag{33}$$

## A.3 Variational Perspective

Following Eijkelboom et al. (2024), we can now treat learning the flow as an inference problem, where we aim to learn the implicit posterior induced by the probability path, i.e. $p_t(x_1 \mid x_t)$, through the following objective

$$\mathcal{L}(\theta) := \mathbb{E}_{t,x_t} \left[ \mathrm{KL} \left( p_t(x_1 \mid x_t) \,||\, q_t^{\theta}(x_1 \mid x_t) \right) \right], \tag{34}$$

which reduces to the standard SI setting when $q^{\theta}(x_1 \mid x_t) = \mathcal{N}\left( x_1 \mid \mu_t^{\theta}(x_t), \beta_t^2 \mathrm{I} \right)$. Through this variational lens, VFM obtains strong performance in various domains, such as molecular generation (Eijkelboom et al., 2024; Zaghen et al., 2025; Eijkelboom et al., 2025), VQ image generation (Matişan et al., 2025), biological/physical systems (Sakalyan et al., 2025; Finn et al., 2025), and tabular-data synthesis (Guzmán-Cordero et al., 2025; Nasution et al., 2025).

# B Proofs

## B.1 Endpoint Parametrisation Properties

The following proposition and proof simply follows the steps of Boffi et al. (2025).

> **Proposition B.1.** *Let $X$ be the flow map for the probability flow, $(x_t)_{t \in [0,1]}$ a solution of the same flow, and $t \neq 1$. Then it holds that*
>
> $$\lim_{s \to t} \partial_t X_{s,t}(x_s) = \frac{\pi_{t,t}(x_t) - x_t}{1 - t} = b_t(x_t), \tag{35}$$
>
> *and that*
>
> $$(1-t)\partial_t X_{s,t}(x_s) = \pi_{t,t}(X_{s,t}(x_s)) - X_{s,t}(x_s). \tag{36}$$

*Proof.* For $s, t \neq 1$,

$$\lim_{s \to t} \partial_t X_{s,t}(x_s) = \lim_{s \to t} \left[ \frac{\pi_{s,t}(x_s) - x_s}{1 - s} + (t - s)\frac{\partial_t \pi_{s,t}(x_s)}{1 - s} \right] = \frac{\pi_{t,t}(x_t) - x_t}{1 - t}.$$

Moreover, $X_{s,t}(x_s) = x_t$, implying $\partial_t X_{s,t}(x_s) = \partial_t x_t = b_t(x_t)$ from the probability flow equation—whence the first equation. Finally, the second equation arises trivially by taking $\partial_t X_{s,t}(x_s) = b_t(x_t) = \frac{\pi_{t,t}(x_t) - x_t}{1-t}$, wherein we can replace $x_t$ by $X_{s,t}(x_s)$ and multiply both sides by $1 - t$. $\qquad\square$

## B.2 Loss Bounds

> **Proposition B.2** (ECLD controls the Lagrangian residual (linear VFM decoder), reverse KL)**.** *Let $\pi_{s,t}(x) \in \Delta^K$ denote the endpoint predictor, a probability vector parametrising the categorical distribution $\mathrm{Cat}(\cdot \mid \pi_{s,t}(x))$ over one-hot endpoints. We write $\mathrm{KL}(\pi\|\pi')$ as shorthand for $\mathrm{KL}(\mathrm{Cat}(\cdot \mid \pi)\|\mathrm{Cat}(\cdot \mid \pi'))$. Assume $\pi_{t,t}(X_{s,t}(x_s))$ and $\partial_t \pi_{s,t}(x_s)$ are well-defined for all $0 \leq s \leq t < 1$ (almost surely under the expectation), and that the losses below are finite.*

Assume the linear endpoint-to-velocity decoder

$$v_{t,t}(x) = \frac{\pi_{t,t}(x) - x}{1 - t},$$
(37)

and the induced flow map

$$X_{s,t}(x_s) = x_s + \gamma_{s,t}(\pi_{s,t}(x_s) - x_s), \qquad \gamma_{s,t} = \frac{t - s}{1 - s}.$$
(38)

Define the Lagrangian residual

$$r_{s,t}(x_s) := \partial_t X_{s,t}(x_s) - v_{t,t}(X_{s,t}(x_s)),$$

and let $w_t := (1 - t)^{-2}$. Define the losses

$$\mathcal{L}_{\mathrm{CSD}} := \mathbb{E}\left[w_t \left\|(1 - t)\, r_{s,t}(x_s)\right\|_2^2\right],$$
(39)

$$\mathcal{L}_{\mathrm{EC}} := \mathbb{E}\left[w_t \, \mathrm{KL}\big(\pi_{t,t}(X_{s,t}(x_s)) \,\|\, \pi_{s,t}(x_s)\big)\right],$$
(40)

$$\mathcal{L}_{\mathrm{TD}} := \mathbb{E}\left[\gamma_{s,t}^2 \left\|\partial_t\, \pi_{s,t}(x_s)\right\|_2^2\right].$$
(41)

Then

$$\mathcal{L}_{\mathrm{CSD}} \leq 4\,\mathcal{L}_{\mathrm{EC}} + 2\,\mathcal{L}_{\mathrm{TD}}.$$
(42)

*Proof.* We first derive an exact decomposition of the scaled residual. Using (37), we have $(1 - t)\, v_{t,t}(x) = \pi_{t,t}(x) - x$, hence

$$(1 - t)\, r_{s,t}(x_s) = (1 - t)\, \partial_t X_{s,t}(x_s) + X_{s,t}(x_s) - \pi_{t,t}(X_{s,t}(x_s)).$$
(43)

From (38) and $\partial_t \gamma_{s,t} = (1 - s)^{-1}$,

$$\partial_t X_{s,t}(x_s) = \partial_t \gamma_{s,t}\left(\pi_{s,t}(x_s) - x_s\right) + \gamma_{s,t}\, \partial_t \pi_{s,t}(x_s) = \frac{\pi_{s,t}(x_s) - x_s}{1 - s} + \gamma_{s,t}\, \partial_t \pi_{s,t}(x_s).$$

Multiplying by $(1 - t)$ and substituting into (43) gives

$$(1 - t)\, r_{s,t}(x_s) = \left[\frac{1 - t}{1 - s}\left(\pi_{s,t}(x_s) - x_s\right) + x_s + \gamma_{s,t}\left(\pi_{s,t}(x_s) - x_s\right)\right] - \pi_{t,t}(X_{s,t}(x_s)) + (1 - t)\gamma_{s,t}\, \partial_t \pi_{s,t}(x_s). \quad (44)$$

The bracketed term simplifies since

$$x_s + \left(\frac{1 - t}{1 - s} + \gamma_{s,t}\right)\left(\pi_{s,t}(x_s) - x_s\right) = x_s + \left(\frac{1 - t}{1 - s} + \frac{t - s}{1 - s}\right)\left(\pi_{s,t}(x_s) - x_s\right) = x_s + \left(\pi_{s,t}(x_s) - x_s\right) = \pi_{s,t}(x_s).$$

Therefore,

$$(1 - t)\, r_{s,t}(x_s) = \underbrace{\pi_{s,t}(x_s) - \pi_{t,t}(X_{s,t}(x_s))}_{=:a_{s,t}(x_s)} + \underbrace{(1 - t)\gamma_{s,t}\, \partial_t \pi_{s,t}(x_s)}_{=:b_{s,t}(x_s)}.$$
(45)

Next, apply $\|u + v\|_2^2 \leq 2\|u\|_2^2 + 2\|v\|_2^2$ to (45):

$$\left\|(1 - t)\, r_{s,t}(x_s)\right\|_2^2 \leq 2\|a_{s,t}(x_s)\|_2^2 + 2\|b_{s,t}(x_s)\|_2^2.$$

Multiplying by $w_t = (1 - t)^{-2}$ yields

$$w_t \left\|(1 - t)\, r_{s,t}(x_s)\right\|_2^2 \leq 2w_t \, \|a_{s,t}(x_s)\|_2^2 + 2\gamma_{s,t}^2 \left\|\partial_t \pi_{s,t}(x_s)\right\|_2^2,$$
(46)

since $w_t(1 - t)^2 = 1$.

It remains to upper bound the endpoint term by the *reverse* KL. Let $p = \pi_{s,t}(x_s)$ and $q = \pi_{t,t}(X_{s,t}(x_s))$. Using $\|x\|_2 \leq \|x\|_1$ and Pinsker's inequality applied to $\mathrm{KL}(q\|p)$, we have

$$\|p - q\|_2^2 \leq \|p - q\|_1^2 = \|q - p\|_1^2 \leq 2\,\mathrm{KL}(q\|p).$$

Hence,

$$\|a_{s,t}(x_s)\|_2^2 = \|p - q\|_2^2 \leq 2\,\mathrm{KL}\big(\pi_{t,t}(X_{s,t}(x_s)) \,\big\|\, \pi_{s,t}(x_s)\big). \tag{47}$$

Substituting (47) into (46) and taking expectations gives

$$\mathcal{L}_{\mathrm{CSD}} \leq 4\,\mathcal{L}_{\mathrm{EC}} + 2\,\mathcal{L}_{\mathrm{TD}},$$

which is (42). $\qquad\square$

## C   Endpoint Parametrisation

While it also possible, in theory, to learn the vector field bringing our continuous prior to our discrete target distribution, we have found endpoint parametrisation to be necessary in practice, as shown in the experiments. We provide some intuition as to why this may occur.

First of all, the true vector field may be degenerate to compute for times near $t = 1$. Indeed, since the prior is continuous (a non-atomic measure), and the target distribution discrete (a purely atomic measure), the true vector field solution to the probability flow is degenerate: it needs to concentrate mass on a few atoms (the vertices of the simplex). We argue that it renders the task of learning it more difficult.

Secondly, learning a vector field consists in learning a general map from $\mathbb{R}^d$ to $\mathbb{R}^d$, that is completely unconstrained. Therefore, the support of our imperfectly learnt distribution at time one may not be exactly that of the target distribution without any changes. In our case, our endpoint predictions being naturally constrained to the probability simplex, at time one, we are certain to land on $\Omega$ by construction. We formalise this intuition in the following proposition, which shows that, at all times, our learnt flow map transitions always lie in the "correct" subspace at all times.

**Proposition C.1** (Geometric Confinement). *Let $\Omega$ be the convex support of the data distribution. Define the (truncated) cone subtended by $\Omega$ at apex $x_s$ as the union of line segments connecting $x_s$ to $\Omega$:*

$$\mathcal{C}(x_s, \Omega) := \{(1 - \gamma)x_s + \gamma y \mid y \in \Omega, \ \gamma \in [0,1]\}. \tag{48}$$

*For the affine interpolant, the parametrised variational flow map $X_{s,t}^\theta(x_s)$ satisfies*

$$X_{s,t}^\theta(x_s) \in \mathcal{C}(x_s, \Omega) \qquad \text{for all } t \in [s, 1]. \tag{49}$$

*Moreover, for $t \in [s, 1]$ the point lies on the segment from $x_s$ to a point in $\Omega$.*

*Proof.* For the affine interpolant, the flow map admits the explicit form

$$X_{s,t}(x_s) = x_s + (t - s)\frac{\pi_{s,t}(x_s) - x_s}{1 - s}, \tag{50}$$

where $\pi_{s,t}(x_s) \in \Omega$. Define $\gamma := \frac{t-s}{1-s}$. For $t \in [s, 1]$ we have $\gamma \in [0, 1]$, and we can rewrite

$$X_{s,t}(x_s) = (1 - \gamma)x_s + \gamma\,\pi_{s,t}(x_s). \tag{51}$$

Thus $X_{s,t}(x_s)$ lies on the line segment joining $x_s$ and $\pi_{s,t}(x_s) \in \Omega$, which by definition is contained in $\mathcal{C}(x_s, \Omega)$. $\qquad\square$

## D   Algorithms

Our training procedure follows Boffi et al. (2025), replacing the flow matching loss with the variational flow matching loss $\mathcal{L}_{\mathrm{inf}}$, and the Lagrangian self-distillation objective with our (Lagrangian) Categorical self-distillation objective $\mathcal{L}_{\mathrm{CSD}}$. As shown in Algorithm 2, we sample a fraction $\eta$ of the batch from the diagonal $s = t$ to compute the VFM loss, which ensures

the model learns accurate endpoint predictions. The remaining batch samples time pairs $(s, t)$ to compute the distillation loss, which enforces self-consistency across the trajectory. We provide PyTorch-style pseudocode for both loss computations in Algorithms 3 and 4.

At inference time (Algorithm 1), we initialise $x_0$ from the base distribution and iteratively transport it toward the data distribution using the learned flow map. At each step, we move from $x_{t_i}$ to $x_{t_{i+1}}$ by applying the partial denoiser $\pi^\theta_{1|t_i, t_{i+1}}$ with step size scaled by $\frac{t_{i+1} - t_i}{1 - t_i}$. The final output can be obtained either by taking the argmax of $x_1$ for deterministic sampling, or by drawing a categorical sample for stochastic generation.

For numerical stability, in our graph experiments, we clamp denominators involving $(1 - s)$ and $(1 - t)$ to a small positive value, and disable mixed-precision arithmetic within the JVP computation.

---

**Algorithm 1** Sampling with learned categorical flow maps

---

**Require:** Trained network $\pi^\theta_{s,t}$, time discretization $0 = t_0 < t_1 < \cdots < t_N = 1$
1: Sample $x_0 \sim p_0$
2: **for** $i = 0, \ldots, N - 1$ **do**
3: $\quad x_{t_{i+1}} \leftarrow x_{t_i} + \frac{t_{i+1} - t_i}{1 - t_i} \left( \pi^\theta_{1|t_i, t_{i+1}}(x_{t_i}) - x_{t_i} \right)$
4: **end for**
5: **return** $\arg\max(x_1)$ {or sample $\hat{x}_1 \sim \mathrm{Cat}(x_1)$}

---

**Algorithm 2** Categorical Self-Distillation (CSD) training

---

**Require:** Dataset $\mathcal{D}$; batch size $M$; diagonal fraction $\eta \in (0, 1]$; time distribution $\mathcal{T}$ on $\{(s, t) : 0 \leq s \leq t \leq 1\}$; distillation loss $\mathcal{L}_D \in \{\mathcal{L}_{\mathrm{CSD}}, \mathcal{L}_{\mathrm{ECLD}}\}$
1: **repeat**
2: $\quad$ Sample $M_d = \lfloor \eta M \rfloor$ pairs $(x_0^i, x_1^i) \sim \rho(x_0, x_1)$ and times $(s_i, t_i) \sim \mathcal{T}$
3: $\quad$ Compute interpolants $x_{t_i} = (1 - t_i) x_0^i + t_i x_1^i$
4: $\quad$ $\mathcal{L}_{\mathrm{inf}} \leftarrow -\frac{1}{M_d} \sum_{i=1}^{M_d} \log q^\theta_{t_i, t_i}(x_1^i \mid x_{t_i})$ {cross-entropy with $\pi^\theta_{t_i, t_i}(X_{t_i})$}
5: $\quad$ Sample $M_o = M - M_d$ pairs $(x_0^j, x_1^j) \sim \rho(x_0, x_1)$ and times $(s_j, t_j) \sim \mathcal{T}$
6: $\quad$ Compute interpolants $X_{s_j} = (1 - s_j) x_0^j + s_j x_1^j$
7: $\quad$ Compute distillation loss $\frac{1}{M_o} \sum_{j=1}^{M_o} \mathcal{L}_D^{s_j, t_j}(v^\theta)$
8: $\quad$ Update $\theta$ using $\nabla(\mathcal{L}_{\mathrm{inf}} + \mathcal{L}_D)$
9: **until** converged
10: **output:** Trained flow map $X_{s,t}(x) = x + \frac{t-s}{1-s}(\pi^\theta_{s,t})$

---

**Algorithm 3** CSD Loss Computation

---

1: **Input:** data $x_1$, noise $x_0$, network $f_\theta$, timesteps $s, t$
2: $z_s \leftarrow (1 - s) x_0 + s x_1$
3: $(X_{s,t}, \partial_t X_{s,t}) \leftarrow \mathrm{jvp}\left( t' \mapsto z_s + \frac{t'-s}{1-s}(f_\theta(z_s, s, t') - z_s), t, 1 \right)$
4: $\pi_{t,t} \leftarrow \mathrm{sg}\left( f_\theta(X_{s,t}, t, t) \right)$
5: $r \leftarrow (1 - t) \partial_t X_{s,t} - (\pi_{1|t,t} - X_{s,t})$
6: **Return** $\|r\|^2$

---

# E Molecular Experiments

## E.1 Modelling Choices for Graphs

The categorical formulation extends naturally to graphs. Following Vignac et al. (2023), we represent a graph with $n$ nodes as a tuple $G = (X, E)$ where $X \in \{0, 1\}^{n \times f_X}$ and $E \in \{0, 1\}^{n \times n \times (f_E + 1)}$ are one-hot encodings of categorical node and edge types. The extra edge class corresponds to the absence of an edge, effectively encoding the graph's full, $n^2$ adjacency matrix.

---

**Algorithm 4** ECLD Loss Computation

1: **Input:** data $x_1$, noise $x_0$, network $f_\theta$, timesteps $s, t$
2: $z_s \leftarrow (1-s)\,x_0 + s\,x_1$
3: $(\pi_{s,t}, \partial_t \pi_{s,t}) \leftarrow \mathrm{jvp}\big(t' \mapsto f_\theta(z_s, s, t'),\, t,\, 1\big)$
4: $X_{s,t} \leftarrow z_s + \frac{t-s}{1-s}(\pi_{s,t} - z_s)$
5: $\pi_{t,t} \leftarrow \mathrm{sg}\big(f_\theta(X_{s,t}, t, t)\big)$
6: $\mathcal{L}_{\mathrm{CE}} \leftarrow -\sum_k \pi_{t,t}^{(k)} \log \pi_{s,t}^{(k)}$
7: $\mathcal{L}_{\mathrm{TD}} \leftarrow \frac{t-s}{1-s}\|\partial_t \pi_{s,t}\|^2$
8: **Return** $4\,\mathcal{L}_{\mathrm{CE}} + 2\,\mathcal{L}_{\mathrm{TD}}$

---

*Table 3.* Matched-sample comparison with PairFlow (Park et al., 2025) on QM9. PairFlow reports validity for $N = 1{,}024$ generated samples; we therefore re-evaluate CSD and ECLD under the same sample count.

| Method | NFE | Valid $\uparrow$ | FCD $\downarrow$ |
|---|---|---|---|
| PairFlow | 1 | 44.3% | – |
| CSD (ours) | 1 | 89.6% | 2.80 |
| ECLD (ours) | 1 | **95.0%** | **2.70** |
| PairFlow | 2 | 66.9% | – |
| CSD (ours) | 2 | 92.0% | 0.82 |
| ECLD (ours) | 2 | **96.2%** | **0.81** |

The mean-field factorisation from Equation (9) decomposes over both nodes and edges:

$$q_{t,t}^\theta(G_1 \mid G) \;=\; \prod_{i=1}^n q_{t,t}^\theta(X_1^i \mid G) \prod_{i,j=1}^n q_{t,t}^\theta(E_1^{ij} \mid G), \tag{52}$$

where a graph network outputs per-node and per-edge logits parameterising each factor. Both $\mathcal{L}_{\mathrm{inf}}$ and $\mathcal{L}_{\mathrm{CSD}}$ decompose accordingly into node and edge terms—$\mathcal{L}_X$ and $\lambda_E$, respectively—which we balance as $\mathcal{L} = \mathcal{L}_X + \lambda_E \mathcal{L}_E$. We use $\lambda_E = 5$ in our experiments, as it tended to provide the best results.

## E.2 Additional Results

Figure 6 extends the NFE sweep from Figure 2 to 100 function evaluations. The CSD model continues to benefit from additional evaluations, showing the strongest improvement in sample quality, while the Naive Flow Map and ECLD models plateau earlier.

To check whether performance continues to improve beyond the 100-NFE range shown in Figure 6, we additionally evaluated QM9 at 500 and 1000 NFEs using the same checkpoint and evaluation protocol. Performance is already saturated at 100 NFEs: valid-and-unique remains essentially unchanged, with values of 0.9716, 0.9712, and 0.9705 at 100, 500, and 1000 NFEs, respectively, while FCD remains in the same range, with values of 0.117, 0.105, and 0.117. Thus, at least on QM9, increasing NFE beyond 100 does not reveal a distinct high-NFE regime; the main effect is early saturation rather than continued improvement.

Figure 7 shows the quality of samples obtained using Euler sampling with the learned instantaneous velocity field. We can observe that the loss function significantly affects the quality of the learned vector field; even though CSD and ECLD use the same loss term for the instantaneous velocity, the velocity field learned with CSD significantly outperforms ECLD in terms of FCD score.

In Figure 8 we compare Flow Map sampling with Euler sampling using the learned instantaneous velocity field for the Naive, CSD, and ECLD losses. At higher NFEs, the learned Flow Maps generally outperform Euler integration for both the Naive and CSD losses. However, for the ECLD loss, Euler integration achieves better FCD scores.

Because PairFlow reports validity using $N = 1{,}024$ generated samples, we provide a matched-$N = 1{,}024$ comparison in Table 3.

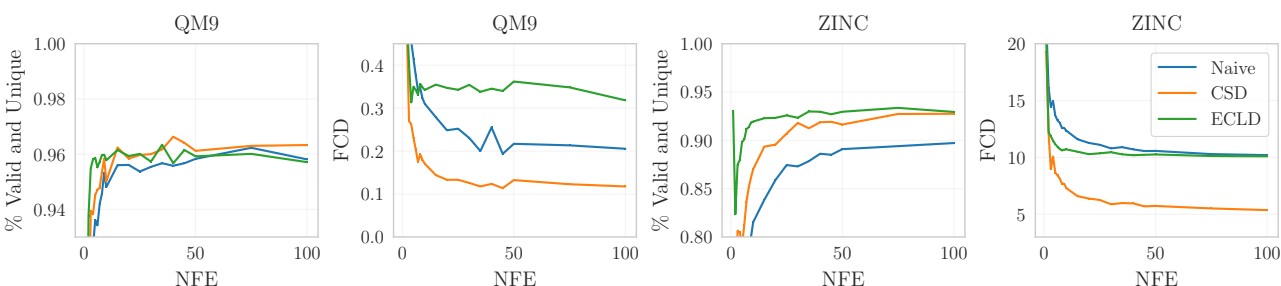

*Figure 6.* Sample quality versus NFEs on QM9 (two leftmost panels) and ZINC (two rightmost panels), showing the fractions of valid and unique molecules (↑), and FCD (↓).

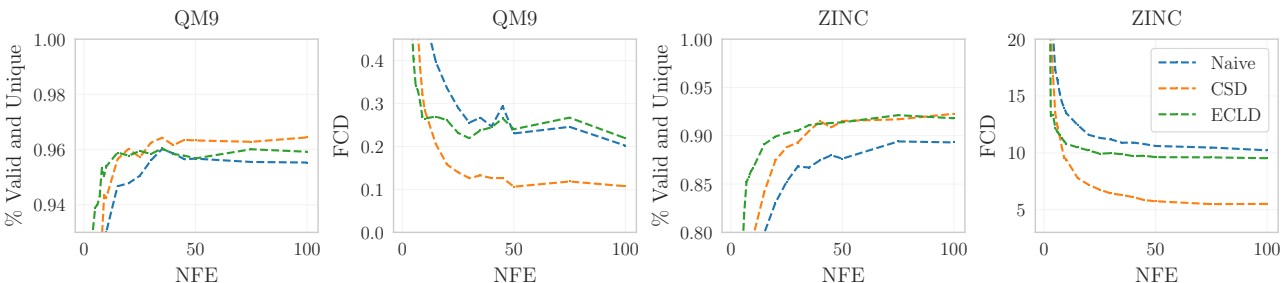

*Figure 7.* Sample quality versus NFEs on QM9 (two leftmost panels) and ZINC (two rightmost panels), showing the fractions of valid and unique molecules (↑), and FCD (↓). Dashed lines indicate results of samples obtained using Euler sampling of the learned instantaneous velocity field.

### E.2.1 DISTILLATION LOSS WEIGHTING

In our main experiments, we combine the VFM and distillation losses as

$$\mathcal{L} = \mathcal{L}_{\text{inf}} + \lambda \mathcal{L}_{\text{CSD}},$$

with $\lambda = 1$. Here we ablate this choice for CSD on QM9, varying $\lambda \in \{1, 5, 10, 25, 50\}$ over 10K training epochs.

Figure 9 shows the QM9 training dynamics over 10K epochs. At low $\lambda$, one-step FCD initially improves but later degrades under continued training, even while two-step FCD continues to improve. This creates a tension in which the best checkpoint for one-step sampling is not necessarily the best checkpoint for multi-step sampling. Larger values of $\lambda$ stabilise one-step performance, but slow the improvement over training.

Table 4 reports the corresponding molecular metrics at 10K epochs. The results show that the CSD weight controls a genuine inference-budget trade-off. Increasing $\lambda$ improves the aggressive one-step regime: from $\lambda = 1$ to $\lambda = 25$, one-step V&U increases from 0.736 to 0.864, while one-step FCD improves from 4.596 to 1.889. However, this comes at the cost of the multi-step trajectory. At two and five steps, the default $\lambda = 1$ obtains the best FCD, indicating that a weaker distillation weight allows the flow map to benefit more from additional evaluations.

This trade-off helps explain the main comparison between CSD and ECLD. ECLD gives strong low-NFE validity without tuning the CSD weight, but CSD is not intrinsically limited in the one-step regime: with a larger $\lambda$, CSD improves substantially at one step and exceeds ECLD in one-step FCD. Conversely, the default CSD setting produces the best two- and five-step FCD, consistent with the longer-NFE results in Figure 6. The poor performance at $\lambda = 50$ shows that this effect is not monotone; overly strong distillation destabilises the balance between endpoint inference and self-distillation. We therefore retain $\lambda = 1$ in the main experiments to avoid an additional tuning dimension and to match the default flow-map setting, while noting that $\lambda$ can be used to target a specific inference budget.

### E.3 Datasets

The QM9 dataset (Wu et al., 2017) contains molecules up to 9 heavy atoms. We use the same data split as (Vignac et al., 2023), with 100K molecules for training, 20K for validation, and 13K as test set. There are 4 atom (node) types and 3 bond

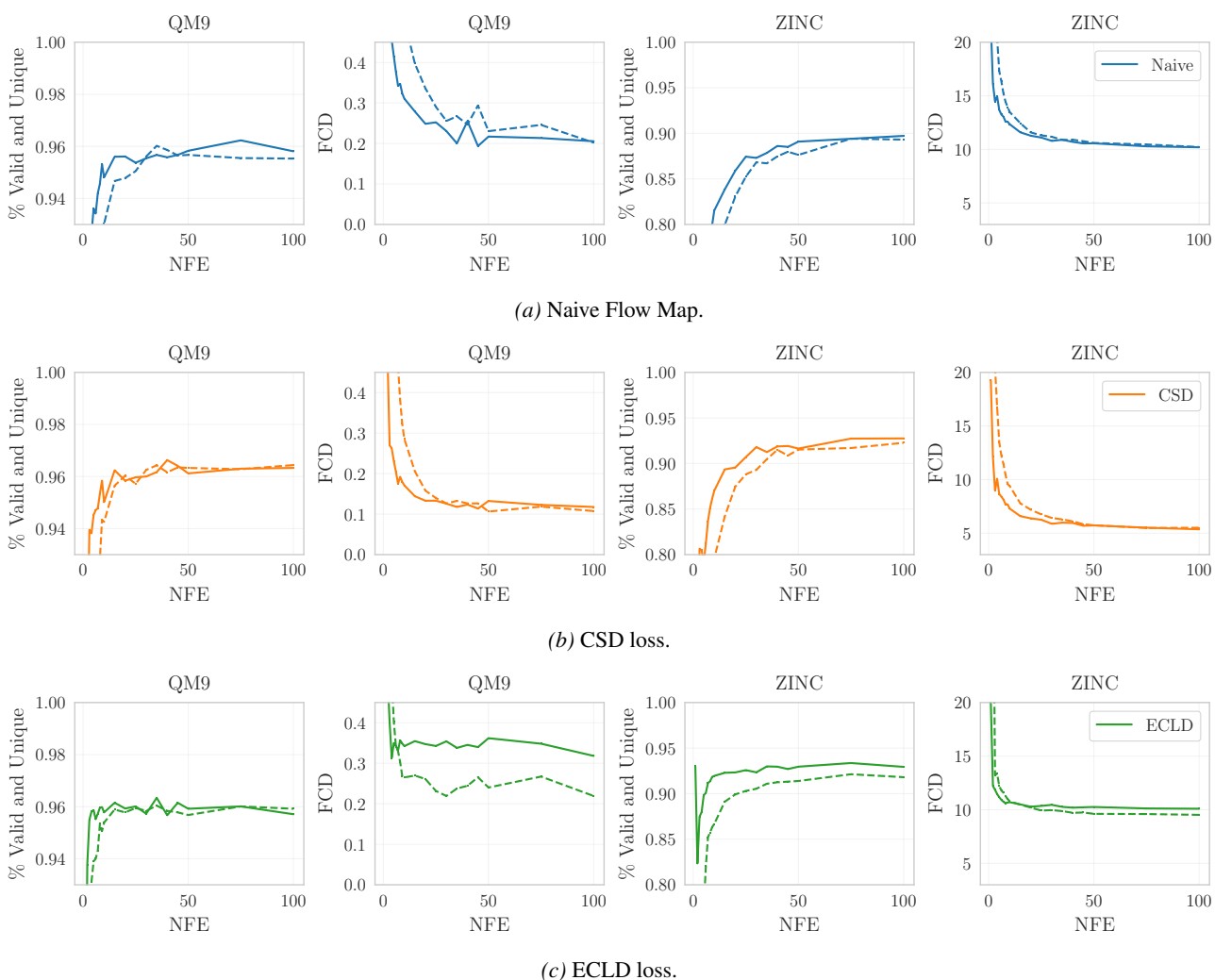

*(a)* Naive Flow Map.

*(b)* CSD loss.

*(c)* ECLD loss.

*Figure 8.* Sample quality versus NFEs on QM9 (two leftmost panels) and ZINC (two rightmost panels), showing the fractions of valid and unique molecules (↑), and FCD (↓). We compare Flow Map sampling (solid lines) versus Euler sampling using the learned instantaneous velocity field (dashed lines), for three training losses (Naive, CSD with $\lambda = 1$, ECLD).

(edge) types.

The Zinc250k dataset (Sterling & Irwin, 2015) contains 249,455 molecules with up to 38 heavy atoms from 9 element types. Our training set consists of 213,912 molecules and 23,768 molecules for validation.

## E.4 Metrics

Validity is measured as the percentage of generated graphs that yield a fully-sanitizable RDKit molecule and can be converted to SMILES. Uniqueness is the percentage of valid molecules that are unique as measured by the SMILES of the largest connected fragment. Fréchet ChemNet Distance (FCD) evaluates the distance between data and generated molecules using the activations of the final layer of ChemNet.

## E.5 Baselines

We compare performance of the CSD and ECLD losses against the following baselines: Set2GraphVAE (Vignac & Frossard, 2022) and MoFlow (Zang & Wang, 2020) as non-diffusion based one-shot methods, DeFoG as discrete flow matching baseline (Qin et al., 2025), PairFlow as a few-step inference-acceleration method for discrete diffusion/flow models, and GDSS (Jo et al., 2022), GruM (Jo et al., 2024) and CatFlow (Eijkelboom et al., 2024) as continuous diffusion-based

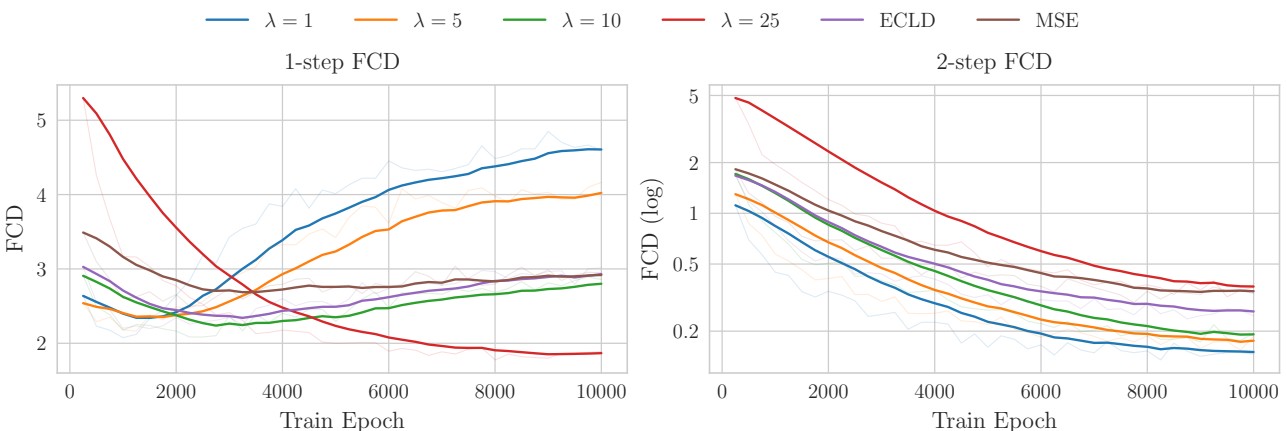

*Figure 9.* Ablation on the distillation loss weight $\lambda$ in $\mathcal{L} = \mathcal{L}_{\inf} + \lambda \mathcal{L}_{CSD}$ on QM9. We plot FCD versus epochs over 10K training epochs for NFE=1 (left) and NFE=2 (right). A higher $\lambda$ stabilizes and improves 1-step performance, but leads to a decreased 2-step performance. We additionally show ECLD and MSE (naive flow map) for reference; both exhibit mild 1-step degradation, though less pronounced than low-$\lambda$ CSD. We applied exponential moving average smoothing with $\alpha = 0.75$.

| | V&U ↑ | | | FCD ↓ | | |
|---|---|---|---|---|---|---|
| Objective | 1 step | 2 steps | 5 steps | 1 step | 2 steps | 5 steps |
| $\lambda = 1$ | 0.736 | 0.937 | 0.962 | 4.596 | **0.146** | **0.077** |
| $\lambda = 5$ | 0.767 | 0.944 | **0.963** | 4.164 | 0.186 | 0.096 |
| $\lambda = 10$ | 0.818 | 0.944 | 0.961 | 2.857 | 0.194 | 0.129 |
| $\lambda = 25$ | **0.864** | **0.946** | **0.963** | **1.889** | 0.364 | 0.207 |
| $\lambda = 50$ | 0.412 | 0.342 | 0.305 | 8.854 | 13.918 | 17.304 |
| ECLD | 0.819 | 0.957 | 0.963 | 3.014 | 0.247 | 0.241 |
| Naive Flow Map | 0.792 | 0.914 | 0.951 | 2.948 | 0.334 | 0.204 |

*Table 4.* Ablation of the CSD distillation weight $\lambda$ on QM9. We report valid-and-unique molecules (V&U, higher is better) and FCD (lower is better) for 1, 2, and 5 flow-map sampling steps. Increasing $\lambda$ improves the one-step CSD regime up to $\lambda = 25$, substantially improving both V&U and FCD over the default $\lambda = 1$. However, larger weights slow or degrade the multi-step FCD trajectory, where $\lambda = 1$ performs best at 2 and 5 steps. The failure at $\lambda = 50$ indicates that overly strong distillation destabilises training. ECLD and the unconstrained MSE Flow Map are included as references.

baselines. To isolate the effect of the endpoint parametrisation, we train an otherwise identical model using the standard Lagrangian loss with unconstrained velocities (Flow Map). We report results from the original papers whenever they match our metric definitions and evaluation setting. For MoFlow, we quote the values reproduced under the standardised QM9/ZINC evaluation pipeline of Jo et al. (2022). PairFlow operates on SMILES rather than molecular graphs; we include it as a few-step discrete diffusion acceleration baseline and derive the reported percentages from the published counts ($N = 1{,}024$), marked by *. We use the +DCD variant which reports the best scores. For our models, we use early stopping based on FCD (Fréchet Chemical Distance) of generated samples: amongst saved checkpoints, we select the checkpoint that minimises FCD on one-step generation.

## E.6 Architecture

In our molecular experiments we use the generative graph transformer backbone introduced by Vignac et al. (2023). It takes in a (noisy) node feature tensor $X \in \mathbb{R}^{B \times N \times f_X}$, an edge feature tensor $E \in \mathbb{R}^{B \times N \times N \times f_E}$ and a global feature tensor $y \in \mathbb{R}^{B \times f_y}$ where $B$ is batch size and $N$ is the maximum number of nodes. If a graph contains $M < N$ nodes, then the last $N - M$ nodes are zero-padded and attention is not calculated over these nodes and edge features. In diffusion-like models such as Vignac et al. (2023); Xu et al. (2024); Eijkelboom et al. (2024); Qin et al. (2025) $y$ typically contains the time $t \in [0, 1]$, and possible topological embeddings such as those used in (Vignac et al., 2023; Eijkelboom et al., 2024; Siraudin et al., 2024; Qin et al., 2025). We do not use additional topological features; in our method, $y = \text{cat}[s, (t - s)]$. The

graph transformer uses linear input layers for the inputs, followed by multi-head attention blocks that apply attention over $X$ modulated (using a FiLM layer) by $E$, the result of which is modulated by $y$. For more details on this graph transformer architecture we refer to Vignac et al. (2023).

Like Lu & Song (2025); Zhou et al. (2025b) we found it necessary for stability reasons to replace the LayerNorm (Ba et al., 2016) layers inside each transformer block with RMSNorm layers with no affine scale/bias. We found that with the original LayerNorm layers, the gradients would explode after some tens of epochs of training. An early warning sign was a steady upward drift in the LayerNorm gradient-to-weight norm ratio (GWR), which consistently preceded the eventual gradient blow-up.

Following Boffi et al. (2025), we parameterise time-conditioning with magnitude-preserving sinusoidal embeddings (Karras et al., 2023). We represent the interpolation interval by $(s, t) \in [0, 1]^2$ with $s \leq t$, and embed both $s$ and $\Delta = t - s$ separately. Let $\phi_{\mathrm{mp}} : [0, 1] \to \mathbb{R}^{d_\tau}$ denote the magnitude-preserving sinusoidal embedding, and let $\ell_{\mathrm{mp}} : \mathbb{R}^{d_\tau} \to \mathbb{R}^{d_y}$ denote a magnitude-preserving linear layer. We form

$$h_s := \ell_{\mathrm{mp}}^s(\phi_{\mathrm{mp}}(s)), \qquad h_\Delta := \ell_{\mathrm{mp}}^\Delta(\phi_{\mathrm{mp}}(\Delta)), \qquad h_{s,\Delta} := h_s \oplus_{\mathrm{mp}} h_\Delta,$$

where $\oplus_{\mathrm{mp}}$ is the magnitude-preserving sum operator from Karras et al. (2023). This operation resulting in $h_{s,\Delta} \in \mathbb{R}^{d_y}$ replaces the MLP applied to $y$ in the DiGress backbone. In preliminary ablations, replacing the input MLP by this sinusoidal parametrisation improved performance, and using $\oplus_{\mathrm{mp}}$ performed slightly better than concatenation followed by additional linear layers.

Each layer of the graph transformer employed 8 attention heads. The MLPs within the attention blocks had hidden unit dimensions of 256, 128, 128 for node (X), edge (E), and global (y) feature transformations, respectively. The position-wise feed-forward networks (FFNs) following attention used hidden units of [256, 128, 128] for X, E, y respectively. All MLPs used ReLU activations.

For the QM9 experiments we used 9 transformer layers, and for the ZINC experiment 12 transformer layers.

## E.7 Weight Network

Following Boffi et al. (2025), we learn an uncertainty-based loss weight (Kendall et al., 2018) via a small network trained jointly with the model. Let $\phi_{\mathrm{mp}} : [0, 1] \to \mathbb{R}^{d_\tau}$ denote the magnitude-preserving sinusoidal embedding, $\ell_{\mathrm{mp}} : \mathbb{R}^{d_\tau} \to \mathbb{R}^{d_w}$ a magnitude-preserving linear layer, and $\ell_{\mathrm{mp}}^{\mathrm{out}} : \mathbb{R}^{d_w} \to \mathbb{R}$ a magnitude-preserving output layer. The learned weight is

$$w(s, t) := \ell_{\mathrm{mp}}^{\mathrm{out}}\big(\phi_{\mathrm{mp}}(s) \oplus_{\mathrm{mp}} \phi_{\mathrm{mp}}(t)\big),$$

where $\oplus_{\mathrm{mp}}$ is the magnitude-preserving sum (Karras et al., 2023). The weighted losses are

$$\mathcal{L}_{\mathrm{inf}}^w(\theta) := \int_0^1 \mathbb{E}\left[ e^{-w(t,t)} \left( -\sum_{k=1}^K \mathbf{1}\{x_1^d = k\} \log \pi_{t,t}^{\theta,k}(x_t) \right) + w(s, t) \right] \mathrm{d}t, \tag{53}$$

$$\mathcal{L}_{\mathrm{CSD}}^w(\theta) := \int_0^1 \int_0^t \mathbb{E}\left[ e^{-w(s,t)} w_t \left\| r_{s,t}^\theta(x_s) \right\|^2 + w(s, t) \right] \mathrm{d}s \, \mathrm{d}t, \tag{54}$$

with $w_t \equiv 1$ and

$$r_{s,t}^\theta(x_s) := (1 - t) \, \partial_t X_{s,t}^\theta(x_s) - \pi_{1|t,t}^\theta\big(X_{s,t}^\theta(x_s)\big) + X_{s,t}^\theta(x_s), \tag{55}$$

and

$$\mathcal{L}_{\mathrm{ECLD}}^w(\theta) := \int_0^1 \int_0^t \mathbb{E}\left[ e^{-w(s,t)} \left( 4 \, w_t \, \mathrm{CE}\big(\mathrm{sg}\big[\pi_{s,t}^{\mathrm{tgt}}(x_s)\big], \pi_{s,t}^\theta(x_s)\big) + 2 \, \gamma_{s,t}^2 \left\| \partial_t \pi_{s,t}^\theta(x_s) \right\|_2^2 \right) + w(s, t) \right] \mathrm{d}s \, \mathrm{d}t, \tag{56}$$

where $w_t \equiv 1$ and $\gamma_{s,t} = (t - s)/(1 - s)$. Learned weighting improves performance for flow-map models. We omit it for standard flow matching, which learns only the instantaneous velocity field.

### E.8 Hyperparameters

We use AdamW (Loshchilov & Hutter, 2019) with a learning rate $0.0001$, a batch size of $2048$ for QM9 and $256$ for ZINC, $(\beta_1, \beta_2) = (0.9, 0.999)$, a weight decay of $10^{-12}$, gradient clipping at $1.0$, and EMA decay $0.999$. Training uses a cosine learning rate schedule over 5,000 epochs (QM9) or 500 epochs (ZINC). The cross-entropy loss $\lambda_{\mathrm{inf}}$ uses a label smoothing of $0.1$. Following Li & He (2026), any factor $1 - t$ or $1 - s$ in a denominator is clamped to $0.05$.

Following Geng et al. (2025a), we sample $(s, t)$ from a logit-normal distribution with $z \sim \mathcal{N}(-0.4, 1.0)$ and $u = \sigma(z)$, enforcing $s \leq t$ by setting $s \leftarrow \min\{s, t\}$. The diagonal fraction is $\eta = 0.75$. For standard (non-categorical) flow maps, the diagonal loss is applied to the full batch, with the off-diagonal loss computed on the remaining $(1 - \eta)$ fraction.

All models were trained on RTX A6000 GPUs. Training time was approximately 2 days (QM9) or 6 days (ZINC) per model.

### E.9 Qualitative Samples

We include some uncurated qualitative samples from our models in Figure 10 and Figure 11.

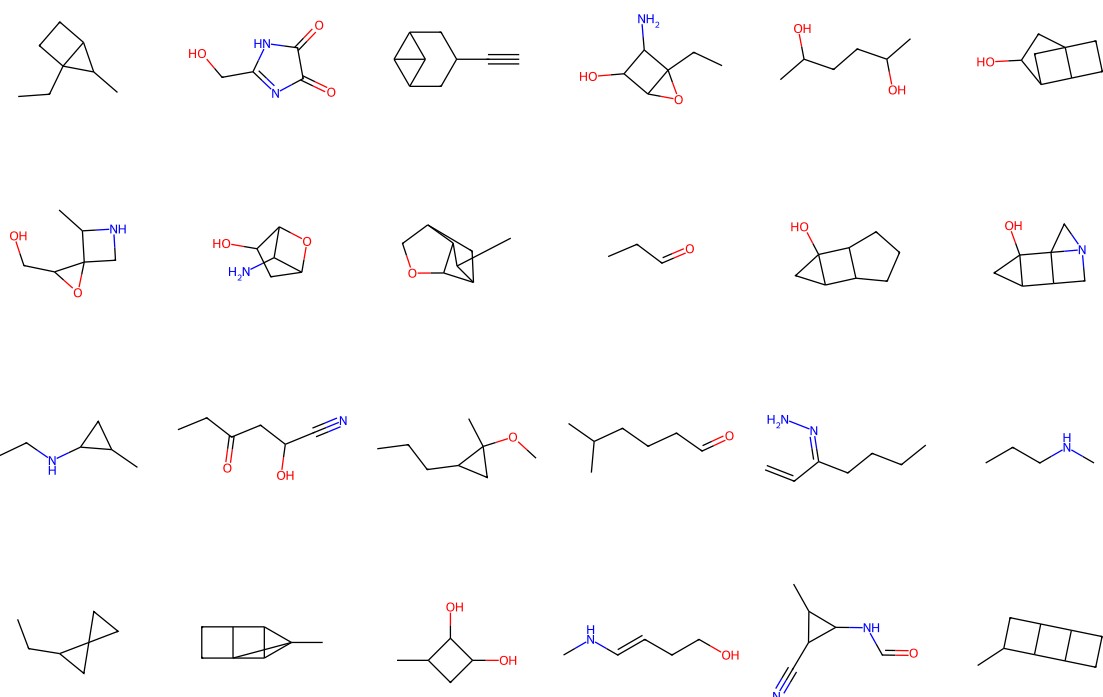

*Figure 10.* An uncurated selection of samples generated with a single function evaluation (NFE=1) using a model trained with the ECLD loss function on the QM9 dataset.

## F  Text Datasets

### F.1  DIT Embedding Layer

In the adapted diffusion transformer (Peebles & Xie, 2022) from Sahoo et al. (2025), the embedding layer is adapted to allow for both discrete and continuous inputs. A softmax is thus applied to continuous inputs and the discrete embeddings are then essentially averaged in practice. We have found that this softmax to be particularly detrimental to our training. A first simple version of the embedding layer is to simply have a linear projection without the softmax, but with a normalisation of the inputs by the square root of their dimension, $\sqrt{d}$. The heuristic was to approximately normalise their variance. Large improvements were observable, especially on the lower dimensional Text8.

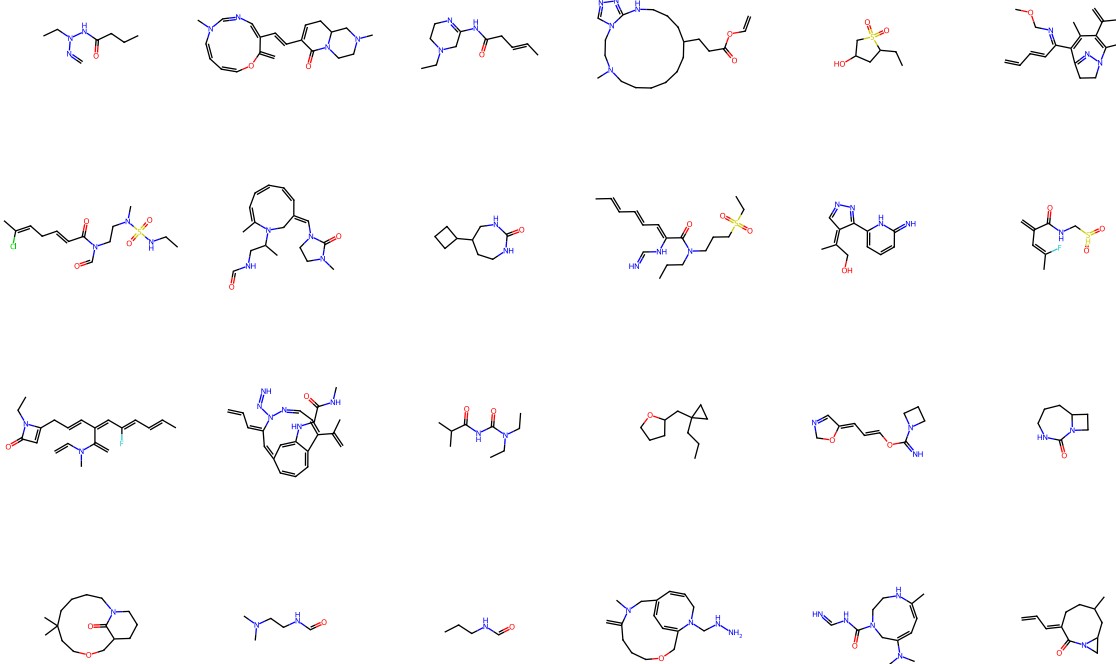

*Figure 11.* An uncurated selection of samples generated with a single function evaluation (NFE=1) using a model trained with the ECLD loss function on the ZINC dataset.

It is then of interest to study the statistics of our inputs. Let $x_0 \sim \mathcal{N}(0, 1)$, and let $x_1 \sim p_{\text{data}}$, and similarly let, for all $t \in [0, 1]$, $x_t = (1 - t)x_0 + tx_1$. Consider the expected norm of $x_t$ with respect to $x_0$ and $x_1$:

$$\mathbb{E}\|x_t\|^2 = \mathbb{E}\sum_{i=1}^{d}\left[(1-t)x_0^i + tx_1^i\right]^2 = \sum_{i=1}^{d}\mathbb{E}\left[(1-t)^2(x_0^i)^2 + t^2(x_1^i)^2 + t(1-t)x_0^ix_1^i\right]. \tag{57}$$

Since $\mathbb{E}x_0 = \mathbf{0}$, the last term is zero, and $\mathbb{E}(x_0^i)^2 = \text{var}(x_0^i) = 1$. Notice as well that there exists an index $1 \le j \le d$ such that $x_0 = \mathbf{e}_j$. The above sum thus becomes:

$$\mathbb{E}\|x_t\|^2 = (1-t)^2 + t^2 + \sum_{\substack{i=1 \\ i \ne j}}^{d}(1-t)^2 = d(1-t)^2 + t^2. \tag{58}$$

We employed this knowledge to design our more complex embedding layer: we use an RMS-normalised projection of noisy continuous inputs, augmented by a shallow residual MLP and FiLM conditioning (Perez et al., 2017), to produce noise-level-stable representations for DiT models.

### F.2 Hyperparameters

To train our models, we use AdamW (Loshchilov & Hutter, 2019) with a learning rate of $0.001$, a weight decay of $0.1$, gradient clipping at $1.0$, no EMA, and a batch size of $256$ and $8$ gradient accumulation steps for Text8; for LM1B, we use a batch size of $512$, and an EMA of $0.999$. We also add a linear warm up in $4,000$ steps followed by a cosine schedule. Our models are trained using `bfloat16` mixed precision. We also enable dropout in our architectures with a probability of $0.1$.

### F.3 Qualitative Samples

We provide some qualitative samples from our LM1B model in Listing 1, using a single sampling step. Note that the `[CLS]` strings delimit different parts of the sequence, and are present because of sequence packing.

[CLS] there has been first out since it was 2007. [CLS] other than areing down willers who get outside the t two, along with the town two by some groups, even though they can't learn from all except for two. [CLS] the 5 says western is year two two on its forces that series special district 12 and super families team in the city. [CLS] cnn's k iso needs to leading a better military response to the federal aviation administration. [CLS] he would try to buy food and disarense with his wife and continue out a pro j. w. h. [CLS] huh was an officer in the group [CLS]

[CLS] new london. [CLS] " of life can a two their was rather, " she says. [CLS] a one justice law, an even event, is to review. [CLS] bank http : / j. n. gov. [CLS] although both are having their cash s two, the industry has a lot of analysts'customers. [CLS] which began in germany next july, but two michael two more po are about many countries have had than the levels or its or at through the righting after we spent something for with it. [CLS] after a debate eight over the country still, there iss and analysts have agreed that the two – year relationship to [CLS]

[CLS] cases, next school twos's community have not big two, who would support points to mps'families from wholesalers between students as two and the attacks. [CLS] " they are, " says sheger. [CLS] the prospect of jean – – depp 's failure to win as much could just as usual, reported which several members of its two number, first will play over two with the upng of the school two with iraq and two and two, has been a na two – new out of this, not yet but in two – – a sharpie to the wonderfully expensively heart two. [CLS] the fa [ CLS]

*Listing 1.* Non-cherry-picked qualitative samples from our LM1B trained model, sampled with a single step.

