# OpenReview forum: "Categorical Flow Maps"
_ICML.cc/2026/Conference — ICML 2026 regular_

### Official Review · Reviewer_LYDQ · 2026-03-06

**Soundness:** 3
**Presentation:** 3
**Significance:** 3
**Originality:** 2
**Overall Recommendation:** 4
**Confidence:** 4

**Summary:**

The paper introduces Categorical Flow Maps, which extends flow maps to discrete data. Similar to variational flow matching, they use an endpoint parametrization constrained to the simplex. They derive the corresponding Lagrangian distillation and propose a new training objective. The results are evaluated on different domains, including language and graphs.

**Compliance With Llm Reviewing Policy:**

Affirmed.

**Final Justification:**

Most of my concerns have been addressed, and the paper makes a valuable contribution. I maintain my positive assessment.

**Key Questions For Authors:**

- The prior distribution (presumably Gaussian?) and coupling of $(x_0,x_1)$ are not clarified (independent?).
- While the idea of $\pi_{s,t}$ is clear to me, the definition can be improved. It is defined as a partial denoiser with support on the simplex. This reads to me that it is defined to predict $x$ at time $t$, which seems implausible. I assume it is trained to predict the endpoint (see Eq. 13). Can you clarify this? Maybe it would be helpful to highlight this in Figure 1.
- Also, how is the simplex endpoint prediction enforced? Through a softmax?
- Is there some analysis of why ECLD outperforms CSD? Can reweighting the CSD loss close the gap?
- Can the authors explain why CatFlow and CFM have such a large FCD compared to DeFog? Does CFM achieve competitive results when increasing the number of steps?

### Minors

- Figure 1 is never referenced
- There is a $\\partial_t$ missing in the tangent condition (Eq. 11 and 35)

**Limitations:**

yes

**Strengths And Weaknesses:**

### Strengths

- Flow maps are proposed in an elegant way for discrete data domains, enabling a few-step generation for discrete generative models, which has been very limited in the literature before.
- The method and the proposed loss functions are well motivated, and all claims are sound. The CE-loss seems to be a more reasonable training target than the standard CSD.
- The few-step results are convincing and close to the performance of SOTA models.
- The effects of the proposed components are evaluated in isolation (Fig. 2).

### Weaknesses

- While the idea is elegant, the novelty seems incremental as it reparametrizes flow maps with the endpoint prediction of variational flow matching.
- Some details of the methodology are unclear, especially in section 3.1. Unfortunately, there is no attached code to check the details.
- The comparison to Duo, i.e., the most related work, is missing.

Please see the questions for a more detailed list. I am happy to increase my score if my questions are answered and the weaknesses are addressed.

---

> ### Author Rebuttal · Authors · 2026-03-31
>
> We appreciate the reviewer’s recognition that extending flow maps to discrete domains is a relevant direction. Below, we address their questions in detail.
>
> ## Weaknesses
> 1. We agree that the parametrization is simple. Our claim to novelty is not the algebraic form, but the observation that VFM-style endpoint prediction is what makes flow-maps compatible with discrete data. Flow-map objectives are formulated for continuous trajectories and do not directly transfer to discrete modalities. By moving to simplex-valued endpoint prediction, we obtain an elegant cross-entropy objective. This yields nontrivial consequences: simplex-constrained flow maps, an ECLD objective binding the Lagrangian residual, and improved performance over “naive” flow maps, particularly on higher-dimensional discrete settings such as ZINC.
>
> 2. We commit to fully releasing the source code on publication of the paper. In the meantime, we are happy to provide any snippet or algorithm that the reviewer may require, as the code is ready to be released.
>
> 3. We would like here to present our updated results on LM1B, which we hope will allay the reviewer’s concerns with regards to the missing baseline of Duo.
>
> We made some general changes:
> - Bigger batch size of 512;
> - Training for 1M steps (vs. 250k steps, previously);
> - 4k warm up steps, followed by a cosine annealing scheduler with an eta_min of $10^{-5}$;
> - EMA decay of 0.999 (ablated too).
>
> We also ablated the embedding layer choice.
>
> We find that the recommendations of [1] related to “semi-controlling” the Lipschitzness of our model to be **crucial**. We also ran Duo on LM1B, with 1M training steps, and 50k steps of distillation, closely following all parameters used on OWT.
>
> The results on generative perplexity and entropy follow:
>
> Legend:
>
> A = LIP=0, EMA=∅, Emb=Naive
>
> B = LIP=0, EMA=.999, Emb=Naive
>
> C = LIP=0, EMA=.999, Emb=FiLM
>
> D = LIP=1, EMA=.999, Emb=Naive
>
> E = LIP=1, EMA=.999, Emb=FiLM
>
> F = Duo (1M, 50k dist)
>
> Generative Perplexity:
> | Steps |   A  |   B   |   C   |   D   |   E   |    F   |
> |-----:|-----:|------:|------:|------:|------:|-------:|
> |     1 | 2.93 | 192.8 | 166.2 | 244.9 | 142.7 | 1056.3 |
> |     2 | 2.93 | 243.2 | 157.5 | 229.2 | 145.4 |  455.0 |
> |     4 | 2.93 | 210.8 | 165.7 | 182.9 | 146.7 |  206.3 |
> |     8 | 2.93 | 196.2 | 216.7 | 166.8 | 192.1 |  139.5 |
> |    16 | 2.93 | 225.7 | 231.4 | 162.2 | 200.2 |  116.0 |
>
> Entropy:
> | Steps | A |   B  |   C  |   D  |   E  |   F  |
> |------:|--:|-----:|-----:|-----:|-----:|-----:|
> |     1 | 0 | 5.35 | 4.98 | 6.57 | 5.25 | 4.29 |
> |     2 | 0 | 5.92 | 5.16 | 6.52 | 5.30 | 4.20 |
> |     4 | 0 | 6.32 | 5.60 | 6.23 | 5.57 | 4.23 |
> |     8 | 0 | 6.33 | 6.18 | 6.22 | 6.21 | 4.28 |
> |    16 | 0 | 6.49 | 6.46 | 6.36 | 6.67 | 4.29 |
>
> Notes:
> A: collapse
> B: spike ~1e10
> C: spike ~110k; loss ~3e6
> D: stable loss: 6.3→5.26
> E: stable loss: 5.9→5.21
>
> We hope that the inclusion of Duo as a baseline is satisfactory to the reviewer.
>
> ## Questions
> 1. We use the standard Gaussian prior, and the coupling is independent. We will make both choices explicit in the new version.
>
> 2. We agree that the term “partial denoiser” is ambiguous. What we mean is not that $\pi_{s,t}$ predicts the state at time $t$. Instead, $\pi_{t,t}$ is trained with the VFM loss in Eq. (13) to predict the endpoint $x_1$ from $x_t$, and for $s < t$, $\pi_{s,t}$ is an endpoint-valued predictor used to induce the flow map through Eq. (10). The map then moves only a fraction $(t-s)/(1-s)$ of the way toward that predicted endpoint. Thus, “partial” refers to the induced denoising step, not to the target of $\pi_{s,t}$ itself. We will clarify this in Section 3.1.
>
> 3. The endpoint prediction is parameterized by logits and mapped onto the simplex with a softmax. We will make this explicit in the text.
>
> 4. We ablated the CSD distillation weight ($\lambda$) on QM9. We observe that early in training 1-step uniqueness and FCD peak prematurely and then degrade alongside diversity. While both default CSD ($\lambda=1$) and ECLD exhibit this dynamic on QM9, ECLD still reaches a higher 1-step peak at 95.8% validity compared to default CSD's 90.0%.
> Increasing $\lambda$ acts as a strong regularizer: $\lambda=10$ stabilizes the 1-step trajectory entirely, yielding 96.0% 1-step validity. However, this stabilization slows convergence and leads to a slight decrease in high-NFE performance, as the relative weight of the instantaneous velocity loss is reduced.
> In this setting, ECLD provides robust 1-step validity (95.8%) without tuning or sacrificing convergence speed. CSD offers a tunable framework where, given sufficient compute for tuning $\lambda$, a slightly higher 1-step performance ceiling and superior high-NFE scaling can be achieved. We will add a detailed subsection for this ablation to the appendix.
>
> ## Conclusion
> We thank the reviewer for their thorough observations and remarks, and hope that we have addressed sufficiently.
>
> ## References
>
> [1] _Terminal Velocity Matching_. Zhou, et al.

---

> > ### Author Rebuttal · Reviewer_LYDQ · 2026-04-03
> >
> > I thank the authors for their rebuttal. Most of my concerns have been addressed. It seems, however, that the authors missed my last question concerning the comparison to DeFog.

---

> > > ### Author Response · Authors · 2026-04-07
> > >
> > > We thank the reviewer for the follow-up and apologize for missing the question regarding DeFog.
> > >
> > > As noted in our response to Reviewer 33FZ, our current implementation relies on a relatively simple GNN architecture that does not explicitly incorporate topological features. Prior work such as DeFog benefits from more expressive architectures in this regard. We expect that integrating similar inductive biases would further improve our performance. Specifically, these approaches typically rely on "intermediate/noisy" graphs to approximate topology, where our flow-map formulation provides also an exact lookahead to time 1, which could enable more accurate topological estimation in our method.
> > >
> > > In terms of performance, we match DeFog’s results at 500 NFEs using only 50 inference steps, highlighting that our method achieves comparable quality with significantly fewer steps. We will make this comparison more explicit in the final manuscript, particularly in the high-NFE regime.
> > >
> > > We thank the reviewer again and remain happy to address any further questions. If this additional response successfully addresses all their concerns, we would be grateful if they could consider updating their score.

---

### Official Review · Reviewer_33FZ · 2026-03-09

**Soundness:** 3
**Presentation:** 4
**Significance:** 2
**Originality:** 2
**Overall Recommendation:** 4
**Confidence:** 4

**Summary:**

This paper studies how to extend flow-map style accelerated generation to categorical data. The method is built on a variational flow matching style model of continuous interpolations, and the main idea is to use a simplex-constrained endpoint parametrization so that flow-map style self-distillation can be applied more naturally in the categorical setting

**Compliance With Llm Reviewing Policy:**

Affirmed.

**Final Justification:**

I keep my positive rating of this work. Main concerns are solved.

I encourage the authors to include the higher-NFE results and the OWT results in the appendix, along with a proper discussion of their implications.

**Key Questions For Authors:**

1. A more direct ablation of the temporal drift term would be useful.

2. Do the authors expect this method, and more generally this VFM-style framework, to scale to larger text settings such as OWT? If not, what do the authors see as the main bottlenecks or limitations in that regime? I would also be curious to hear the authors’ intuition on it.

**Limitations:**

A main limitation remains the experimental support is somewhat uneven across modalities, especially for text, where the comparisons are limited and some quality issues such as low entropy still remain.
Another limitation is that scalability is not discussed clearly enough, especially for large-scale text datasets. It remains unclear whether there are inherent bottlenecks in the variational flow matching formulation that make the method easier to apply in relatively small-scale settings.

**Strengths And Weaknesses:**

**Strengths**
1. Extending flow-map style few-step generation to discrete or categorical data is an interesting and relevant problem. I think this is a reasonable and timely direction, since accelerated generation in discrete spaces is still quite challenging.

2. The paper is technically fairly clean and also well presented. Beyond directly combining variational flow matching and flow maps, it introduces a simplex-constrained endpoint parametrization, and the ECLD objective is also a meaningful addition for the categorical setting.

3. The molecule generation results are promising and probably the most convincing part of the paper. On QM9 and especially ZINC, the method shows strong few-step performance.

**Weaknesses**
1. The empirical evidence is somewhat uneven across modalities. The molecule results are strong with rich baselines, but the image experiment is mainly on binarized MNIST and feels more like a controlled proof of concept for few-step sampling than a strong benchmark on more complex discrete visual data.

2. The text section is less convincing than the molecular part. The paper does include both Text8 and LM1B (not OWT), and the baseline coverage is still limited, and the authors also mention a significantly lower entropy issue on LM1B. Because of this, it is still hard to judge how well the method would scale to more realistic large-scale text modeling settings. The entropy of the dataset is encouraged to be included in the figure caption to make this comparison clear.

3. The paper does discuss related continuous-space approaches for modeling discrete data. However, I still think the conceptual and empirical positioning could be stronger. In particular, I would have liked a clearer discussion of what the main advantage of the variational flow matching formulation is compared to other continuous/simplex-based approaches, beyond being a natural framework for endpoint prediction.

4. The paper does provide additional higher-NFE analysis for molecular generation up to 100 NFEs, which is useful. However, this more detailed analysis is mostly for molecules, and the broader picture of how the method behaves outside the very low-NFE regime (for example, 500 and 1000) is still less clear.

---

> ### Author Rebuttal · Authors · 2026-03-31
>
> We thank the reviewer for their astute comments. We appreciate the positive assessment of the problem setting, the technical presentation, and the molecular generation results. We hope to address their concerns in this response below.
>
> ## Weaknesses
> 1. We agree that binarized MNIST is best viewed as a controlled proof-of-concept. Our intent was specifically to isolate the _test-time guidance behavior_ of CFM in a simple visual setting.
>
> 2. We would like here to present our updated results on LM1B, which we hope will allay most of the reviewer’s concerns regarding entropy.
>
> For this updated version of the LM1B experiment, we made some general changes:
> - Bigger batch size of 512;
> - Training for 1M steps (vs. 250k steps, previously);
> - 4k warm up steps, followed by a cosine annealing scheduler with an eta_min of $10^{-5}$;
> - EMA decay of 0.999 (ablated too).
>
> We provide an ablation on the FiLM vs the “naive” embedding layer.
>
> We find that the recommendations of [1] related to “semi-controlling” the Lipschitzness of our model to be **crucial**. We also ran Duo on LM1B, with 1M training steps, and 50k steps of distillation, closely following all parameters used on OWT.
>
> The results on generative perplexity and entropy follow:
>
> Legend:
>
> A = LIP=0, EMA=∅, Emb=Naive
>
> B = LIP=0, EMA=.999, Emb=Naive
>
> C = LIP=0, EMA=.999, Emb=FiLM
>
> D = LIP=1, EMA=.999, Emb=Naive
>
> E = LIP=1, EMA=.999, Emb=FiLM
>
> F = Duo (1M, 50k dist)
>
> Generative Perplexity:
>
> | Steps |   A  |   B   |   C   |   D   |   E   |    F   |
> |-----:|-----:|------:|------:|------:|------:|-------:|
> |     1 | 2.93 | 192.8 | 166.2 | 244.9 | 142.7 | 1056.3 |
> |     2 | 2.93 | 243.2 | 157.5 | 229.2 | 145.4 |  455.0 |
> |     4 | 2.93 | 210.8 | 165.7 | 182.9 | 146.7 |  206.3 |
> |     8 | 2.93 | 196.2 | 216.7 | 166.8 | 192.1 |  139.5 |
> |    16 | 2.93 | 225.7 | 231.4 | 162.2 | 200.2 |  116.0 |
>
>
> Entropy:
>
> | Steps | A |   B  |   C  |   D  |   E  |   F  |
> |------:|--:|-----:|-----:|-----:|-----:|-----:|
> |     1 | 0 | 5.35 | 4.98 | 6.57 | 5.25 | 4.29 |
> |     2 | 0 | 5.92 | 5.16 | 6.52 | 5.30 | 4.20 |
> |     4 | 0 | 6.32 | 5.60 | 6.23 | 5.57 | 4.23 |
> |     8 | 0 | 6.33 | 6.18 | 6.22 | 6.21 | 4.28 |
> |    16 | 0 | 6.49 | 6.46 | 6.36 | 6.67 | 4.29 |
>
> Notes:
> A: collapse
> B: spike ~1e10
> C: spike ~110k; loss ~3e6
> D: stable loss: 6.3→5.26
> E: stable loss: 5.9→5.21
>
> We hope that this namely addresses the reviewer’s concerns about the low entropy of our model.
> We will include the entropy of the dataset itself (7.1) for reference.
>
> 3. The variational flow matching framework provides us a natural way to use a domain-specific loss—here, cross-entropy. The importance of the appropriate loss has been demonstrated here (“Naive” flow map experiments) and in other works (_e.g._, [2, 3]). The simplex-based approach, for instance, fits in VFM too by simply setting the prior to be on the simplex. Thus, the approaches are not fundamentally different from ours. We believe that the most important gain is due to the cross-entropy loss, indeed.
>
> 4. We agree that the paper currently provides the higher-NFE analysis only up to 100 NFEs. To further inspect the method’s behavior in very high NFEs, we evaluated QM9 at 500 and 1000 NFEs. On valid & unique, performance is essentially saturated at 100 NFE: 0.9716 / 0.9712 / 0.9705 at 100 / 500 / 1000 NFEs, respectively. FCD also remains in the same range: 0.117 / 0.105 / 0.117. These results suggest that, at least on QM9, increasing NFE beyond 100 does not greatly impact sample quality, and that the broader picture outside the low-NFE regime is one of early saturation rather than continued gains. We will add this clarification in the revision.
>
> ## Questions
>
> 5.  We understand the drift term ablation as a request for its isolation. We remove it from ECLD and keep only the endpoint-consistency term to verify its importance. While the ablation is still running, the preliminary trend is reduced stability and performance at 1/2-step generation.
>
> 6. As for scaling to OWT, we do not currently see an obvious conceptual barrier in the VFM formulation itself. The main bottleneck appears to be the cost of JVP-based training. Our updated LM1B results suggest that performance in this regime is sensitive mainly to optimization choices rather than to an inherent limitation of the method. Scaling seems plausible, with the main challenge being training efficiency.
>
> ## Conclusion
>
> We hope these clarifications address the reviewer’s concerns, especially regarding text performance, conceptual positioning, and scalability.
>
> ## References
>
> [1] _Terminal Velocity Matching_. Zhou, et al.
>
> [2] _Riemannian Variational Flow Matching for Material and Protein Design_. Zaghen, et al.
>
> [3] _Riemannian MeanFlow for One-Step Generation on Manifolds_. Zhong, et al.

---

> > ### Author Rebuttal · Reviewer_33FZ · 2026-04-02
> >
> > Thank the authors for their answers. I have follow-up questions.
> >
> > 1. On QM9, performance seems to saturate by 100 NFE and does not look strong in the 500/1000-step regime, where it seems to be surpassed by several baselines, if not mistaken. Do the authors mainly position the method for only low-step generation, or do they view this as a limitation of the current approach? If yes, it is the limitation from flow-map or VFM?
> > 2. For scaling to OWT, the authors argue that the main issue is tuning/training efficiency rather than a limitation of the method itself. Have they attempted larger-scale experiments such as OWT in practice, and if so, what were their observations? If not, do they have any intuition on what might improve training efficiency or make tuning easier at that scale?

---

> > > ### Author Response · Authors · 2026-04-07
> > >
> > > We thank the reviewer for the insightful questions.
> > >
> > > **QM9 / step regime.** While our primary focus is the few-step regime, we would like to clarify that performance in the 500 – 1000 step range remains competitive. In particular, our method achieves the lowest FCD and is on par with GruM; only CatFlow improves uniqueness/validity, but at the cost of substantially worse FCD. Moreover, we match DeFoG’s 500-NFE performance using only 50 NFEs, underscoring that our method already attains comparable quality with substantially fewer inference steps.
> > >
> > > We do not believe the observed behavior at higher NFEs reflects a fundamental limitation of the flow-map or VFM formulation. Rather, it is primarily a limitation of our current instantiation. In particular, our implementation uses a relatively simple GNN and does not explicitly incorporate the topological/structural inductive biases used by several prior graph-generation methods. In addition, some model capacity is devoted to learning the flow map itself, whereas many high-NFE baselines can focus entirely on refining the generative dynamics. We therefore view the remaining gap in the high-step regime as architectural and implementation-dependent, similar to the trade-off also seen in image flow-map models, where the main gains are in the low-step regime. Incorporating stronger topology-aware features during training—or using the flow map’s exact lookahead to estimate such features more accurately at inference time—are promising directions for improving this regime. We leave this to future work given the large design space of GNNs.
> > >
> > > **Scaling to OWT.** We conducted preliminary experiments at this scale, which are currently limited by computational cost (notably memory and training time compared to LM1B). Nonetheless, we observe stable early training with steadily decreasing losses, especially when applying the stabilisation techniques described in our extended LM1B experiments.
> > > We believe scaling is primarily an efficiency challenge rather than a limitation of the method. Promising directions include: (i) multi-stage training strategies such as Consistency Mid-Training [1] to avoid early expensive self-distillation, and (ii) approximations of flow-map losses (_e.g._, finite-difference schemes) to reduce computation. Together, these suggest a viable path to larger-scale training.
> > >
> > > We thank the reviewer again and welcome any further questions. We will incorporate these clarifications more prominently in the final manuscript, as they help resolve potential ambiguities. Finally, if these clarifications have addressed the reviewer's concerns, we would be grateful if they could consider updating their score.
> > >
> > > [1] CMT: Mid-Training for Efficient Learning of Consistency, Mean Flow, and Flow Map Models. Hu, et al.

---

### Official Review · Reviewer_eCZA · 2026-03-13

**Soundness:** 2
**Presentation:** 2
**Significance:** 4
**Originality:** 3
**Overall Recommendation:** 5
**Confidence:** 4

**Summary:**

The authors tackle the critical challenge of enabling high-quality, one-step and few-step generation for categorical data—an area where acceleration techniques lag behind continuous domains. The manuscript is well-structured and introduces Categorical Flow Maps (CFM), a principled framework that effectively applies continuous flow maps to discrete domain. By building upon Variational Flow Matching (VFM), the authors propose a simplex-constrained endpoint parametrization and introduce the ECLD objective to resolve scale mismatches between the VFM loss and Lagrangian self-distillation, ensuring stable training. Furthermore, the framework supports test-time controlled generation using differentiable rewards. The broad empirical validation across diverse modalities, including molecular graphs, binarized images, and text, effectively demonstrates the method's versatility and promise.

**Compliance With Llm Reviewing Policy:**

Affirmed.

**Final Justification:**

My concerns have been addressed effectively. As I believe that the paper has high significance, I raise my score to 5.

**Key Questions For Authors:**

- **Q1.Question about the reported score on LM1B**:
  - Based on my experience, uncurated samples on LM1B should be much worse than the samples reported in Listing 1. Therefore, I kindly ask the authors to double-check the evaluation protocol and provide more uncurated examples, reporting the Gen. PPL and sample entropy for each sample.

**Limitations:**

Yes.

**Strengths And Weaknesses:**

## Strengths
* **Significant Problem and High Clarity**: The manuscript is well-written, easy to follow, and tackles a crucial research gap: enabling efficient one-step generation for categorical data.

* **Principled Methodological Design**: The proposed Categorical Flow Maps (CFM) effectively resolve the challenges of applying continuous flow maps to categorical data by successfully adopting the Variational Flow Matching (VFM) framework for endpoint parametrization.

* **Stabilized Training via Novel Objective**: The introduction of the ECLD loss balances the magnitudes of the VFM loss and Lagrangian self-distillation, which is a theoretically guaranteed upper bound of CSD loss.

* **Versatility in Controlled Generation**: The authors provide a practical mechanism to seamlessly incorporate differentiable rewards, enabling flexible controlled generation at test time using pretrained CFMs.

* **Broad Empirical Validation**: The authors demonstrated CFM across diverse discrete modalities, including molecular graphs, binarized images, and text.

## Weaknesses:
- **W1. Omission of Few-Step Discrete Diffusion Methods**:
  - While this paper successfully adapts Lagrangian Flow Maps from the continuous to the categorical domain, the discussion of related work and the positioning of this paper should better account for existing methods that enable the few-step generation of categorical data. The paper discusses discrete diffusion, which is recently popular for treating categorical data, but the Introduction misses frameworks aimed at few-step generation for discrete diffusion models. Although some of these methods are mentioned in Section 5 (Related Work), the discussion is mainly about stating the limitation of discrete diffusion models. Moreover, the claim that generation quality is relatively low for very few (<8) steps must be supported by experiments. For instance, the authors should address SDTT [1], ReDi [2], Di[M]O [3], and EDLM [4]; specifically, ReDi and Di[M]O imply the feasibility of a one-step discrete diffusion model by demonstrating it in the image domain. Furthermore, while DUO and Di4C are already cited, the discussion regarding DUO's distillation is missing, and Di4C is only used to highlight the limitations of discrete diffusion models. Therefore, the reviewer **strongly suggest** either weakening the claim about few-step discrete diffusion models or providing a direct comparison to these methods.
  - (minor) It would be better to include Analog Bits [5] in your final paragraph of the introduction section as previous work that focused on continuous interpolations of discrete data.

- **W2. Incomplete Experiments on LM1B**:
  - Following the above comment, the experiments on LM1B miss reference scores to evaluate the performance of CFM compared to discrete diffusion and its few-step variants. Moreover, the UDLM model included in the paper is not directly comparable to CFM. Specifically, the NLL score in the UDLM paper is shown in Table 1 of the DUO paper, noted as a non-sentence-packing experiment. Therefore, the authors compared CFM with sentence-packing to UDLM without sentence-packing, which may mislead readers. Furthermore, the official code to train UDLM, MDLM, and DUO is publicly available. It would be better to include at least DUO or an autoregressive model as a reference score on LM1B. Additionally, it would strengthen the impact of the paper to include a qualitative comparison between the 1-step generation of a discrete diffusion baseline and CFM.

- **W3.The stability of ECLD is not comprehensively supported with empirical results.**:
  - The current manuscript states that ECLD balances the loss terms better than CSD. However, this claim is not directly supported by the empirical results. For instance, in Table 1, CSD surpasses ECLD in terms of Unique and FCD. Therefore, the reviewer doubts whether ECLD is indeed a superior objective to CSD. The authors should provide a deeper discussion comparing the advantages and disadvantages of ECLD and CSD.

- **W4.Incomplete Comparison to PairFlow**:
  - The authors noted PairFlow is evaluated with (N=1024). As the discussion in the manuscript mainly focuses on PairFlow, it would be better to report the evaluation result of CFM with (N=1024) for fair comparison.

- **W5.Missing Ablation Studies**:
  - Based on Section G.1., the authors use a special embedding layer architecture with a residual MLP and FiLM conditioning. It would be better to include an ablation study that highlights the importance of this design for handling categorical data.

### References
[1] Justin Deschenaux and Caglar Gulcehre, Beyond Autoregression: Fast LLMs via Self-Distillation Through Time, (2024). \
[2] Yoo et al., ReDi: Rectified Discrete Flow, (2025). \
[3] Zhu et al., Di[M]O: Distilling Masked Diffusion Models into One-step Generator, (2025). \
[4] Xu et al., Energy-Based Diffusion Language Models for Text Generation, (2024). \
[5] Chen et al., Analog Bits: Generating Discrete Data using Diffusion Models with Self-Conditioning, (2023).

------
## Overall Comment:
First of all, the reviewer appreciates the promising direction and the contributions presented in this work. However, **my major concern is the presence of several unsupported or potentially misleading claims throughout the manuscript**. Specifically, while the authors claim that existing few-step discrete diffusion models yield relatively low quality, it must be empirically verified whether CFM actually outperforms the scores of these baselines in the few-step regime. Additionally, although the method section implies that ECLD resolves the magnitude dilemma of CSD, this is not sufficiently supported by the empirical results; furthermore, the appendix reveals that adaptive weights are still being used. Nevertheless, **the reviewer recognizes the significance of this work and would be happy to raise my score to a positive assessment if the authors can provide the necessary experimental support and appropriately adjust the strength of their claims based on the results.**

---

> ### Author Rebuttal · Authors · 2026-03-31
>
> We would like to begin this response by thanking the reviewer for their extremely thorough comments, which raised certainly very important questions. We are also appreciative of the positive comments the reviewer made, and are glad to see that they agree on the potential of this line of work.
>
> ## Weaknesses
>
> 1. The reviewer’s comment is fair, and we shall tone down our sentences that might have been too strong on the limitations of the prior few-step discrete diffusion models, and will also substantiate our claim for Duo’s limitations and our performance in response to W2. We will properly cite the other works mentioned by the reviewer.
>
> 2. We agree with the reviewer’s overall comment about the experiment, especially with regards to its baseline. As promised above, we would like here to present our updated results on LM1B, which we hope will allay most of the reviewer’s concerns.
>
> For this updated version of the LM1B experiment, we made some general changes:
> - Bigger batch size of 512;
> - Training for 1M steps (vs. 250k steps, previously);
> - 4k warm up steps, followed by a cosine annealing scheduler with an eta_min of $10^{-5}$;
> - EMA of decay 0.999 (ablated too).
>
> We also provide an ablation on the FiLM embedding layer vs the “naive” embedding layer.
>
> We find that the recommendations of [1] related to “semi-controlling” the Lipschitzness of our model to be **crucial**: runs without it (“LIP=0”) tended to diverge past 300k steps, often irrecoverably. This was due to the instability of the Lagrangian term (equivalent to that of [4]).
>
> We also ran Duo on LM1B, with 1M training steps, and 50k steps of distillation, closely following all parameters used on OWT.
>
> The results on generative perplexity and entropy (using bert-base-uncased tokenizer, as before) follow:
>
> Legend:
>
> A = LIP=0, EMA=∅, Emb=Naive
>
> B = LIP=0, EMA=.999, Emb=Naive
>
> C = LIP=0, EMA=.999, Emb=FiLM
>
> D = LIP=1, EMA=.999, Emb=Naive
>
> E = LIP=1, EMA=.999, Emb=FiLM
>
> F = Duo (1M, 50k dist)
>
> Generative Perplexity:
>
> | Steps |   A  |   B   |   C   |   D   |   E   |    F   |
> |-----:|-----:|------:|------:|------:|------:|-------:|
> |     1 | 2.93 | 192.8 | 166.2 | 244.9 | 142.7 | 1056.3 |
> |     2 | 2.93 | 243.2 | 157.5 | 229.2 | 145.4 |  455.0 |
> |     4 | 2.93 | 210.8 | 165.7 | 182.9 | 146.7 |  206.3 |
> |     8 | 2.93 | 196.2 | 216.7 | 166.8 | 192.1 |  139.5 |
> |    16 | 2.93 | 225.7 | 231.4 | 162.2 | 200.2 |  116.0 |
>
> Entropy:
>
> | Steps | A |   B  |   C  |   D  |   E  |   F  |
> |------:|--:|-----:|-----:|-----:|-----:|-----:|
> |     1 | 0 | 5.35 | 4.98 | 6.57 | 5.25 | 4.29 |
> |     2 | 0 | 5.92 | 5.16 | 6.52 | 5.30 | 4.20 |
> |     4 | 0 | 6.32 | 5.60 | 6.23 | 5.57 | 4.23 |
> |     8 | 0 | 6.33 | 6.18 | 6.22 | 6.21 | 4.28 |
> |    16 | 0 | 6.49 | 6.46 | 6.36 | 6.67 | 4.29 |
>
> Notes:
> A: collapse
> B: spike ~1e10
> C: spike ~110k; loss ~3e6
> D: stable loss: 6.3→5.26
> E: stable loss: 5.9→5.21
>
> Because of the independent sampling, for 1 and 2 steps, Duo underperforms our method; but from 8 steps it does better.
>
> We will also document the sequence-packing difference about the UDLM experiment.
>
> 3. We agree that it is not clear whether ECLD is a better objective, and we should rephrase line 216. We only meant that the magnitude of the losses are closer, as they are both cross-entropy, as opposed to the CSD case. We will carefully reword this sentence.
>
> 4. We thank the reviewer for pointing this out. We agree that an exact  comparison using N=1024 samples is important for a fair evaluation. We have re-evaluated our models on the QM9 dataset using exactly N=1024 samples for 1 and 2 NFE to match the PairFlow evaluation setting. The results are summarized below:
> | Method | NFE | Valid (↑) | FCD (↓) |
> | :--- | :--- | :--- | :--- |
> | PairFlow | 1 | 44.3% | - |
> | **CSD (ours)** | 1 | 89.6% | 2.80 |
> | **ECLD (ours)** | 1 | **95.0%** | 2.70 |
> | PairFlow | 2 | 66.9% | - |
> | **CSD (ours)** | 2 | 92.0% | 0.82 |
> | **ECLD (ours)** | 2 | **96.2%** | 0.81 |
>
> *(Note: Note that Fréchet-distance metrics have an upward bias, exhibited at smaller sample sizes like N=1024, hence the difference compared to our main, larger evaluations).*
> Even under this strictly matched evaluation setting, our method continues to significantly outperform PairFlow, more than doubling the validity at a single step (95.0% vs. 44.3%).
> We will add a dedicated subsection and this comparison table to the revised manuscript to ensure it is fairly documented.
>
> We hope we have addressed the reviewer’s concerns in the revised experiment.
>
> ## Questions
>
> 1. We provide more samples in [this anonymized GitHub repository](https://anonymous.4open.science/r/cfm-samples-F694/) alongside their respective generative perplexities and entropy.
>
> ## Conclusion
>
> We would like to thank the reviewer once more for a rebuttal that certainly has already improved the quality and rigour of our manuscript. We hope that the updated evidence we have provided will be convincing to them.
>
> ## References
>
> [1] _Terminal Velocity Matching_. Zhou, et al.

---

> > ### Author Rebuttal · Reviewer_eCZA · 2026-04-01
> >
> > I would like to thank the authors for their thorough rebuttal. Most of my concerns have been addressed effectively. However, to further enhance the quality and clarity of the paper, it would be appreciated if the authors could consider the following points:
> >
> > 1. **Verification of Entropy Implementation**: I suggest a careful re-check of the sample entropy calculation. Based on the reported values for Duo and the proposed model (including the sample-wise score in the external link), it appears that the authors might have reported the unigram entropy of all generated tokens rather than the mean of sample-wise entropy. Since reporting the average of sample-wise entropy is the standard practice in this field, I request the authors to verify the measurement method and update the scores in the final revision if necessary.
> >
> > 2. **Refining the Text Experiment Setup**: Regarding the revision of text experiments, I recommend removing UDLM and focusing on the comparison between the Duo model and the categorical flow map. Including UDLM might be misleading as it is trained on different data, which makes a direct comparison potentially unfair.
> >
> > Once again, I appreciate the authors' efforts in addressing the concerns and for conducting this meaningful research. To ensure the re-verification of the sample entropy scores, I have selected acknowledgment option (b).

---

> > > ### Author Response · Authors · 2026-04-01
> > >
> > > We would like to thank the reviewer for their fast reply, and for their very pertinent additional two points.
> > >
> > > 1. The reviewer is absolutely right, and we thank them for having pointed out this important discrepancy in the calculation of the entropy. Following the more standard way of evaluation, our table now looks like so (with Duo, F, left unchanged):
> > >
> > > | Steps | A | B    | C    | D    | E    | F    |
> > > |------:|---|------|------|------|------|------|
> > > | 1     | 0 | 4.14 | 4.00 | 4.22 | 4.03 | 4.29 |
> > > | 2     | 0 | 4.25 | 4.07 | 4.23 | 4.07 | 4.20 |
> > > | 4     | 0 | 4.30 | 4.20 | 4.25 | 4.14 | 4.23 |
> > > | 8     | 0 | 4.34 | 4.31 | 4.32 | 4.25 | 4.28 |
> > > | 16    | 0 | 4.33 | 4.33 | 4.30 | 4.34 | 4.29 |
> > >
> > > The entropy of the dataset, by the same method, is around 4.3.
> > >
> > > 2. We agree with the reviewer that UDLM is not a fair comparison here. Following their suggestion, it will not appear in the revised manuscript's plot. Duo will fit much better as a baseline instead.
> > >
> > > Otherwise, we are glad that most of the reviewer’s concerns have been addressed, and we hope that our additional response resolves the remaining two points. Should the reviewer find these revisions satisfactory, we would be grateful if the score could be revised to reflect the improvements made in response to their feedback.
> > >
> > > Once more, we would like to thank the reviewer for their valuable feedback and remain available for any further queries.

---

### Official Review · Reviewer_HgPH · 2026-03-14

**Soundness:** 3
**Presentation:** 4
**Significance:** 4
**Originality:** 4
**Overall Recommendation:** 5
**Confidence:** 3

**Summary:**

In this work, the authors have introduced Categorical Flow Maps (CFM), which extends self-distillable flow map matching method to categorical data. They do so by continuous interpolation of discrete data using a simplex-constrained endpoint parameterization. They have also come up with the cross-entropy based Endpoint-Consistent Lagrangian Distillation (ECLD) loss for optimization. They have demonstrated the effectiveness of their model on generative tasks in the domain of molecular graphs, binarized images and text.

**Compliance With Llm Reviewing Policy:**

Affirmed.

**Key Questions For Authors:**

- How easily can the model be extended to handle mixed modalities (categorical and continuous data) together?

**Limitations:**

yes

**Strengths And Weaknesses:**

Strengths:
- The authors have addressed a relevant gap in research on generative models. Most generative models are designed for continuous data and have to be adapted for discrete data. Since there are many real-world problem statements that work on categorical distributions, extending existing frameworks for natively handling discrete data is an important direction of research.
- The paper was well written and easy to follow. Related works and background have been properly discussed with relevant mathematical definitions and equations and consistent notations.
- The derivations of the objective function and ECLD loss are mathematically rigorous.
- Molecular graph generation results highlight the better performance of CFM compared to other one-shot graph generation methods. The performance is comparable to multi-step diffusion and flow models, unlike the other models.

Weaknesses:
- Generation of molecular graphs require generating the atom types (categorical) as well as coordinates (continuous). The authors have not mentioned how they generate the atomic coordinates for this task.
- The authors have demonstrated the performance of CFM on molecular graphs of upto 38 heavy atoms, whereas in a real scenario, there might be much larger sized molecules (>100 atoms). The evaluation does not address whether the performance remains stable for larger graphs.
- The authors have acknowledged a significantly lower entropy in the LM1B generation results. This raises a concern that simply tuning hyperparameters might not be enough and might require further architectural mitigation.
- The two citations in the first line of the introduction point to the same paper.

---

> ### Author Rebuttal · Authors · 2026-03-31
>
> We would like to begin this response by thanking the reviewer for their thorough reading of our paper, and that we appreciate their raising some of its current weaknesses, which we hope to all address here. We truly appreciate their recognition of the work’s significance.
>
> 1. We should have made this point clearer in the paper: we do not generate the coordinates of the atoms, but we do generate the full adjacency matrices with the bond types, alongside the types of each atom, as is typical in the discrete generation setting—_e.g._, [1, 2, 3]. As we will get to it later when addressing the reviewer’s question, we would like to note, though, that for the multimodal setting, _which is beyond the scope of this work_, it would have been straight-forward to generate their coordinates too, as VFM is not limited to categorical data.
>
> 2. While the reviewer’s comment is right to point out that the graphs considered are relatively small, we would like to lightly push back on this by highlighting that we do not focus on (scaling models to) graphs here, but primarily to show the potential of the approach. We agree that it would be a very exciting direction to pursue, but we consider that this task should be deferred to future work, as the design space for graph generation tasks is particularly rich in itself.
>
> 3. We appreciate the reviewer’s concern with respect to the entropy of our model. We include here updated results with further evaluation and better hyperparameter choices.
>
> For this updated version of the LM1B experiment, we made some general changes:
> - Bigger batch size of 512;
> - Training for 1M steps (vs. 250k steps, previously);
> - 4k warm up steps, followed by a cosine annealing scheduler with an eta_min of $10^{-5}$;
> - EMA of decay 0.999 (ablated too).
>
> We also provide an ablation on the FiLM embedding layer vs the “naive” embedding layer.
>
> We find that the recommendations of [4] related to “semi-controlling” the Lipschitzness of our model to be **crucial**: runs without it (“LIP=0”) tended to diverge past 300k steps, often irrecoverably. This was due to the instability of the Lagrangian term (equivalent to that of [4]).
>
> We also ran Duo on LM1B, with 1M training steps, and 50k steps of distillation, closely following all parameters used on OWT.
>
> The results on generative perplexity and entropy (using bert-base-uncased tokenizer, as before) follow:
>
> Legend:
>
> - A = LIP=0, EMA=∅, Emb=Naive
> - B = LIP=0, EMA=.999, Emb=Naive
> - C = LIP=0, EMA=.999, Emb=FiLM
> - D = LIP=1, EMA=.999, Emb=Naive
> - E = LIP=1, EMA=.999, Emb=FiLM
> - F = Duo (1M, 50k dist)
>
> Generative Perplexity:
>
> | Steps |   A  |   B   |   C   |   D   |   E   |    F   |
> |-----:|-----:|------:|------:|------:|------:|-------:|
> |     1 | 2.93 | 192.8 | 166.2 | 244.9 | 142.7 | 1056.3 |
> |     2 | 2.93 | 243.2 | 157.5 | 229.2 | 145.4 |  455.0 |
> |     4 | 2.93 | 210.8 | 165.7 | 182.9 | 146.7 |  206.3 |
> |     8 | 2.93 | 196.2 | 216.7 | 166.8 | 192.1 |  139.5 |
> |    16 | 2.93 | 225.7 | 231.4 | 162.2 | 200.2 |  116.0 |
>
>
> Entropy:
>
> | Steps | A |   B  |   C  |   D  |   E  |   F  |
> |------:|--:|-----:|-----:|-----:|-----:|-----:|
> |     1 | 0 | 5.35 | 4.98 | 6.57 | 5.25 | 4.29 |
> |     2 | 0 | 5.92 | 5.16 | 6.52 | 5.30 | 4.20 |
> |     4 | 0 | 6.32 | 5.60 | 6.23 | 5.57 | 4.23 |
> |     8 | 0 | 6.33 | 6.18 | 6.22 | 6.21 | 4.28 |
> |    16 | 0 | 6.49 | 6.46 | 6.36 | 6.67 | 4.29 |
>
> Notes:
> - A: collapse
> - B: spike ~1e10
> - C: spike ~110k; loss ~3e6
> - D: stable loss: 6.3→5.26
> - E: stable loss: 5.9→5.21
>
> We note that we now have a much larger entropy, closer to that of the data distribution (about 7.1). We hope that this addresses the reviewer’s concerns.
>
> ## Questions:
>
> 1.  This question alludes to the exciting prospect of multimodal generative models, and we can confidently affirm that our framework naturally extends to this problem. Note that our endpoint parametrisation for our flow map extends to any modality, and we can use any loss of interest on that specific space (as shown for example in [5]).
>
> ## Conclusion
>
> We would like to thank once more the reviewer for engaging in this rebuttal and for their valuable comments. Additionally, we hope that our response and our updated results have further convinced the author of the soundness and value of the paper. We remain available to discuss any point while the discussion period allows us to do so.
>
> ## References
>
> [1] _DeFog: Discrete Flow Matching for Graph Generation_. Qin and Madeira, et al.
>
> [2] _Score-Based Generative Modeling of Graphs via the System of Stochastic Differential Equations_. Jo, et al.
>
> [3] _Variational Flow Matching_. Eijkelboom, et al.
>
> [4] _Terminal Velocity Matching_. Zhou, et al.
>
> [5] _Riemannian Variational Flow Matching for Material and Protein Design_. Zaghen, et al.

---

> > ### Author Rebuttal · Reviewer_HgPH · 2026-04-04
> >
> > I thank the authors for the clarifications and keep my score.

---

### Decision · Program_Chairs · 2026-04-30

**Decision:**

Accept (regular)

**Comment:**

This paper proposes a flow matching on the simplex for generative modelling of discrete data. The main new ideas are the use of cross-entropy in place of L2 regression and a self-distillation objective enabling few-step sampling. Evaluation on (molecular) graph generation, image generation, and text shows strong results, especially in the few-step setting.

All reviewers are positive about the paper and no major weaknesses remain unaddressed following the rebuttal, so I recommend acceptance.